

# RECEPTOR MODELLING OF BOTH PARTICLE COMPOSITION AND SIZE DISTRIBUTION FROM A BACKGROUND SITE IN LONDON, UK – THE TWO STEP APPROACH

## David C.S. Beddows and Roy M. Harrison[*][†]

**National Centre for Atmospheric Science**
**School of Geography, Earth and Environmental Sciences**
**University of Birmingham**
**Edgbaston, Birmingham B15 2TT**
**United Kingdom**

[*]To whom correspondence should be addressed.
Tele: +44 121 414 3494;  Fax: +44 121 414 3708;  Email: r.m.harrison@bham.ac.uk

[†]Also at:  Department of Environmental Sciences / Center of Excellence in Environmental Studies, King Abdulaziz University, PO Box 80203, Jeddah, 21589, Saudi Arabia

**ABSTRACT**

Some air pollution datasets contain multiple variables with a range of measurement units, and combined analysis by Positive Matrix Factorization (PMF) can be problematic, but can offer benefits from the greater information content. In this work, a novel method is devised and the source apportionment of a mixed unit data set ($PM_{10}$ mass and Number Size Distribution NSD) is achieved using a novel two-step approach to PMF. In the first step the $PM_{10}$ data is PMF analysed using a source apportionment approach in order to provide a solution which best describes the environment and conditions considered. The time series G values (and errors) of the $PM_{10}$ solution are then taken forward into the second step where they are combined with the NSD data and analysed in a second PMF analysis. This results in NSD data associated with the apportioned $PM_{10}$ factors. We exemplify this approach using data reported in the study of Beddows et al. (2015), producing one solution which unifies the two separate solutions for $PM_{10}$ and NSD data datasets together. We also show how regression of the NSD size bins and the G time series can be used to elaborate the solution by identifying NSD factors (such as nucleation) not influencing the $PM_{10}$ mass.

**Keywords:** $PM_{10}$; London; PMF; source apportionment; receptor modelling

# 1.    INTRODUCTION

It is unquestionable that worldwide, the scientific vista of air quality is expanding; whether it is the increasing number of observatories or the refinement of information mined from the increasing sophistication of measurements often incorporated in campaign work. The number of metrics being measured has increased from simple measurements of PM mass and gas concentrations, and we can now probe the composition of the PM mass and the size distributions with mass spectrometers, mobility analysers and optical devices.

Studies using PMF as a tool for source apportionment of particle mass using multicomponent chemical analysis data are published frequently using datasets from around the world.  However, they do not always provide consistent outcomes (Pant and Harrison, 2012), and one means by which source resolution and identification can be improved is by inclusion of auxiliary data, such as gaseous pollutants (Thimmaiah et al., 2009), particle number count (Masiol et al., 2017) or particle size distribution (Beddows et al., 2015; Ogulei et al., 2006; Leoni et al., 2018).

Harrison et al. (2011), analysed NSD data (merged SMPS and APS data) with PMF using auxiliary data (meteorology, gas concentration, traffic counts and speed).  The study used particle size distribution data collected at the Marylebone Road supersite in London in the autumn of 2007 and put forward a 10 factor solution comprised of roadside and background particle source factors.  Sowlat et al., 2016 carried out a similar analysis on number size distribution (13nm - 10µm) data combined with several auxiliary variables collected in Los Angeles.  These included BC, EC/OC, PM mass, gaseous pollutants, meteorological, and traffic flow data. A six-factor solution was chosen comprising of: nucleation, 2 x traffic, an urban background aerosol, a secondary aerosol and a soil factor. The two traffic sources contributed up to above 60% of the total number concentrations combined.  Nucleation was

also observed as a major factor (17%). Urban background aerosol, secondary aerosol, and soil, with relative contributions of approximately 12, 2.1, and 1.1%, respectively, overall accounted for approximately 15% of PM number concentrations, although these factors dominated the PM volume and mass concentrations, due mainly to their larger mode diameters. Chan et al. (2011) considered extracting more source information from an aerosol composition dataset by including data on other air pollutants and wind data in the analysis of a small but comprehensive dataset from a 24-hourly sampling programme carried out during June 2001 in an industrial area in Brisbane. They chose multiple types of composition data (aerosols, VOCs and major gaseous pollutants) and wind data in source apportionment of air pollutants and found it to result in better defined source factors and better fit diagnostics, compared to when non-combined data were used. Likewise, Wang et al. (2017) report an improvement in source profiles when coupling the PMF model with $^{14}$C data to constrain the PMF run as *a priori* information.

However, while combining, for example, particle chemical composition and size distribution data in a single PMF analysis may assist source resolution, difficulties arise if the two datasets have different and/or ambiguous rotations (discussed in Section 2). This tends to result in factors with either mass contributions and small number contributions or number contributions and small mass contributions and rarely a meaningful contribution from both data types. Experimental design can of course circumnavigate this problem, for instance, using chemical data which is already size segregated, measured using a cascade impactor (Contini et al., 2014). Such an approach is attractive by view of the fact that there is no question as to whether both datasets sufficiently overlap across the size bins. However, cascade impactors do not offer the high time resolution of particle counting instruments, with individual measurements lasting hours or days. Even so, for the case where two or more

instruments are available in a campaign to measure two or more different metrics, e.g. PM
mass and particle number (PN), then a combined data analysis is useful. Emami and Hopke
(2017) have shown that the effect of adding variables as auxiliary data (with potentially
different units) to a NSD data set is to decrease the rotational ambiguity of a solution from a
1-step PMF analysis.

In this study, we present a method for analysing simultaneously collected $PM_{10}$ composition
and NSD data. In the work of Beddows et al. (2015), both particle composition and number
size distribution (NSD) data from a background site in London (2011 and 2012) was
analysed using Positive Matrix Factorization. As part of the methodology development, it
was concluded that it was preferable not to combine these two data types in a single analysis
but to conduct separate PMF analyses for $PM_{10}$ mass and particle number. This yielded a
6 factor solution for the $PM_{10}$ data (Diffuse Urban; Marine; Secondary; Non-Exhaust
Traffic/Crustal (NET/Crustal)); Fuel Oil; and Traffic. Factors described as Diffuse Urban;
Secondary; and Traffic were identified in the 4 factor solution for the NSD data, together with
a Nucleation factor not seen in the $PM_{10}$ mass data analysis (see Figure 1). When combining
the $PM_{10}$ and NSD data in a single PMF analysis, Diffuse Urban; Nucleation; Secondary;
Aged Marine and Traffic Factors were identified but the factors were not as clearly separated
from each other as the factors derived from the separate datasets. For example, Fuel Oil
was now mixed in with Marine and called Aged Marine. This is summarized in Figure 1.
However, it would still be useful to obtain a number size distribution for each of the 6 $PM_{10}$
factors and/or a chemical composition for the 4 NSD factors. As a continuation of this work,
we present an alternative method for analysing the combined dataset in a so called, two-
step methodology. In the first step, we analyse the mass data ($PM_{10}$; units: $\mu g/m^3$) according
to the methodology of Beddows et al. (2015). This results in a time series factor G which is

carried forward into a second PMF analysis of a combined dataset consisting of the G time series and an auxillary data set (i.e. NSD; units: $1/cm^3$).  The first step identifies sources and apportions the G factors to their contribution to mass and in the second step, an FKEY matrix is chosen such that G 'drives' the model and the NSD data 'follow'.  This means that we have $PM_{10}$ factors each of which is augmented by its number size distribution. Furthermore, we also consider linear regression as a second step in a PMF-LR analysis to show that although the initial analysis is biased toward mass by analysing $PM_{10}$ factors only, unseen factors influencing the NSD data (e.g. nucleation) can be identified in the data.

## 2. EXPERIMENTAL

With a population of 8.5 million in 2014 (ONS, 2017), the UK city of London is the focus of study in this work where the London *North Kensington* (NK) Site ($LAT = 51^o : 31' : 15.780''$ N and $LONG = 0^o : 12' : 48.571'' $ W ) was considered.  NK is part of both the London Air Quality Network and the national Automatic Urban and Rural Network and is owned and part-funded by the Royal Borough of Kensington and Chelsea.  The facility is located within a self contained cabin within the grounds of Sion Manning School. The nearest road, St. Charles Square, is a quiet residential street approximately 5 metres from the monitoring site and the surrounding area is mainly residential.  The nearest heavily trafficked roads are the B450 (~100 m East) and the very busy A40 (~400 m South).  For a detailed overview of the air pollution climate at North Kensington, the reader is referred to Bigi and Harrison (2010).

## 2.1 Data

As alluded to, this work is a continuation of the study carried out by Beddows et al (2015), which analysed NSD and $PM_{10}$ chemical composition data collected at the London NK receptor site.  Number Size Distribution (NSD) data were collected continuously every 15

min using a Scanning Mobility Particle Sizer (SMPS) consisting of a CPC (TSI model 3775) combined with an electrostatic classifier (TSI model 3080) and air dried according to the EUSAAR protocol (Wiedensohler et al., 2012). The particle sizes covered were 51 size bins ranging from 16 nm to 604 nm and the 15 min distributions were aggregated up to hourly averages (where there were at least 3 x 15 min samples per hour) and all missing values were replaced using a value calculated using the method of Polissar et al. (1998). Further details of the SMPS settings are given in Table S1 and the reader is also referred to Beccaceci et al. (2013a,b) for an extensive account of how the NSD data was collected and quality assured.

Accompanying the NSD data from the study of Beddows et al. (2015) was the PMF output from the analysis of $PM_{10}$ chemical composition data. The latter data consisted of 24h air samples taken daily over a 2-year period (2011 and 2012) using a Thermo Partisol 2025 sampler fitted with a $PM_{10}$ size selective inlet. These filters were analysed for total metals $PM_{metals}$ (Al, Ba, Ca, Cd, Cr, Cu, Fe, K, Mg, Mo, Na, Ni, Pb, Sn, Sb, Sr, V, and Zn), using a Perkin Elmer/Sciex ELAN 6100DRC following HF acid digestion of GN-4 Metricel membrane filters. Water-soluble ions $PM_{ions}$ ($Ca^{2+}$, $Mg^{2+}$, K, $NH_4^+$, $Cl^-$, $NO_3^-$ and $SO_4^{2-}$) were measured using a near-real-time URG-9000B (hereafter URG) ambient ion monitor (URG Corp). The data capture over the 2 years ranged from 48 to 100% as different sampling instruments varied in reliability. Data gaps were filled by measurements made on daily $PM_{10}$ filter samples collected continuously at this site using a Partisol 2025; laboratory-based ion chromatography measurements were made for anions on Tissuquartz ™2500 QAT-UP filters) . No cation measurements were available from these filters, and this resulted in a lower data capture for the cations. Again, all missing data were replaced using a value calculated using the method of Polissar et al. (1998). A woodsmoke metric, CWOD, was

also included. This was derived as PM Woodsmoke from the methodology of Sandradewi et al. (2008) utilising Aethalometer and EC/OC data, as described in Fuller et al. (2014). Samples were also collected using a Partisol 2025 with a $PM_{10}$ size selective inlet and concentrations of elemental carbon (EC) and organic carbon (OC) were measured by collection on quartz filters (Tissuquartz ™ 2500 QAT-UP) and analysis using a Sunset Laboratory thermal–optical analyser according to the QUARTZ protocol (which gives results very similar to EUSAAR 2: Cavalli et al., 2010) (NPL, 2013).  We refer to CWOD, EC and OC as $PM_{carbon}$. In addition, particle mass was determined on samples collected on Teflon-coated glass fibre filters (TX40HI20WW) with a Partisol sampler and $PM_{10}$ size-selective inlet.

This aforementioned $PM_{10}$ data was represented in this work as the PMF solution for $PM_{10}$-only data, derived in Beddows et al. (2015) and consisting of 6 sources, namely: Diffuse Urban; Marine; Secondary; Non-Exhaust Traffic/Crustal; Fuel Oil; and Traffic.  The Diffuse Urban factor had a chemical profile indicative of contributions mainly from both woodsmoke (CWOD) and road traffic (Ba, Cu, Fe, Zn).  The Marine factor explained much of the variation in the data for Na, $Cl^-$ and $Mg^{2+}$, and the Secondary factor was identified from a strong association with $NH_4^+$, $NO_3^-$, $SO_4^{2-}$ and organic carbon. For the Traffic emissions, the PM did not simply reflect tailpipe emissions, as it also included contributions from non-exhaust sources, i.e. resuspension of road dust and primary PM emissions from brake, clutch and tyre wear. The Non-Exhaust Traffic/Crustal factor explained a high proportion of the variation in the Al, $Ca^{2+}$ and Ti measurements consistent with particles derived from crustal material, derived either from wind-blown or vehicle-induced resuspension. There was also a significant explanation of the variation in elements such as Zn, Pb, Mn, Fe, Cu and Ba, which had a strong association with non-exhaust traffic emissions. As there was a strong

contribution of crustal material to particles resuspended from traffic this likely reflected the
presence of particulate matter from resuspension and traffic-polluted soils. The last factor
was attributed to Fuel Oil, characterised by a strong association with V and Ni together with
significant $SO_4^{2-}$.  This output comprised the first-step solution in the 2-step analysis of $PM_{10}$
and NSD data and in this study we concentrate on the analysis of the NSD data in the
second PMF step with the aim of assigning a NSD to each of the 6 $PM_{10}$ factors.

**2.2     Methods**
**2.2.1     PMF**
Positive Matrix Factorization (PMF) is a well-established multivariate data analysis method
used in the field of aerosol science.  PMF can be described as a least-squares formulation
of factor analysis developed by Paatero (Paatero and Tapper, 1994). It assumes that the
ambient aerosol concentration $X$ (represented by $n \times m$ matrix of $n$ observations and $m$ $PM_{10}$
constituents or NSD size bins), measured at one or more sites, can be explained by the
product of a source profile matrix $F$ and source contribution matrix $G$ whose elements are
given by equation 1:

$$x_{ij} = \sum_{k=1}^{p} g_{ik} \cdot f_{kj} + e_{ij} \qquad \text{i=1…n; j=1…m} \tag{1}$$

where the $j^{th}$ PM constituent (element, size bin, or auxiliary measurement) on the $i^{th}$
observation (i.e. hour) is represented by $x_{ij}$. The term $g_{ik}$ is the contribution of the $k^{th}$ factor
to the receptor on the $i^{th}$ hour, $f_{kj}$ is the fraction of the $j^{th}$ PM constituent in the $k^{th}$ factor, and
$e_{ij}$ is the residual for the $j^{th}$ measurement on the $i^{th}$ hour. The residuals (i.e. difference
between measured and reconstructed concentrations) are accounted for in matrix $E$ and the
two matrices $G$ and $F$ are obtained by an iterative algorithm which minimises the object
function Q (see equation 2).

Using the data and uncertainty matrices for the model, equation 1 is optimised in the PMF
algorithm by minimising the Q value (equation 2),

$$Q = \sum_{i=1}^{n}\sum_{j=1}^{m}\left(\frac{e_{ij}}{s_{ij}}\right)^2 \tag{2}$$


where $s_{ij}$ is the uncertainty in the $j^{th}$ measurement for hour $i$. All analyses were carried out
in Robust mode which reduces the impact of outliers (Paatero, 2002).

PMF is a weighted technique and the value of Q, and hence the model fit, is determined by
the input variables with the lowest values of uncertainty, $s_{ij}$ , thus giving their variables a
higher weighting in the analysis. Input variables with low weight have little effect upon the
value of Q, even when their residuals are large. This can be used to the advantage of the
operator, e.g. when apportioning total PM mass in a conventional one-step PMF, the total
PM concentrations are normally input with artificially high uncertainty, so that they are
essentially passive in the PMF analysis and do not influence its outcome. By doing so, the
chemical composition data determine the apportionment of PM mass to the source-related
factors identified by the PMF. A similar approach can be followed in the PMF analysis of a
combined dataset where higher weightings can be applied to the main dataset of interest
such that it "drives" the analysis and the auxillary data set "follows", i.e. the uncertainties are
chosen such that the balance of total weights from the two data sets is *tipped* towards the
measurement of interest and highest reliability in regards of rotational unabiguity.
To assess the PMF model, the $Q$ value is outputted by PMF and compared to a theoretical
value $Q_{theory}$ which is approximately the difference between the product of the dimensions of
$X$ and the product of the number of factors and the sum of dimensions of $X$ (i.e. n x m – p(n
+ m)) pk x m.  For a given number of factors, the whole uncertainty matrix is scaled by a
factor $b_{scale}$ until the ratio between Q and $Q_{theory}$ is approximately one (rQ value = Q/$Q_{theory}$ =

244  1 ± 0.02).


With regards to the final output from PMF, a scaling has to be applied in order to achieve
quantitative results.  This is done by scaling either G or F to unity such that the units from X
are carried over to either F or G respectively to complete the apportionment.  However,
different routes have to be considered depending on whether X has homogeneous or
heterogeneous units.

**2.2.2  *1-Step method using data in the same units - homogeneous units***
Given a PMF input data matrix X, a solution GF + E can be computed where G represents
the time series of the source profiles F, with a residual matrix E.  Often X comprises columns
of $PM_{10}$ component concentrations (e.g. ICPMS values measured from acid-digested filters
collected with a Partisol 2025) and it is common practice to also include a Total variable
(e.g. column of $PM_{10}$, measured using a TEOM) in the data matrix.  The resulting $PM_{10}$
profile element value can then be used to scale G and F such that G carries the units of X
with F unitless.  Note that neither G or F is scaled to unity in this approach. Instead, scaling
is done after the analysis using a constant $a_k$, determined by the time series of a Total
variable (e.g. $PM_{10}$), down weighted by applying a high uncertainty, within the input data.

$$x_{ij} = \sum_{k=1}^{p} (a_k g_{ik}) \left(\frac{f_{kj}}{a_k}\right)$$


The resulting value for the $PM_{10}$ contribution for each factor within the F matrix is then used
as a scaling constant $a_k$ in equation 3. Such scaling results in unitless factors F which
describe the characteristics of the sources and time series G with units of $\mu g/m^3$.
Apportionment can then be carried out by averaging the G values for each source factor, or
a fully quantified time series of each factor can be presented, e.g. in Bivariate plots. Of
course, the G and F can be normalized such that G is unitless and F carries units; an
approach necessary when X contains heterogeneous units. This approach however,
requires each column of G to be scaled to unity, by using the PMF setting Mean IGI = 1.

**2.2.3**     *1-Step method using data with different units - heterogeneous units*
If the analysis of X was to be enhanced by the inclusion of data from a second instrument
with different units, then a different approach to the *1-Step method with homogeneous units*
would be required to analyse the joint data matrix [X,Z] = G[X,Z] F[X,Z] + E[X,Z]. If the
previous method was applied where F was normalized, then it would not be clear what units
to assign to G, whether the units from X or Z. To get around this problem, G is scaled to
unity. This results in a unitless time series G and a quantified F matrix. For each source
profile the sum of the species associated with either data type gives the average total
apportionment, e.g. of $PM_{10}$ or number concentration PN. Of course, this requires the
complete mass or number closure of the elements making up either $PM_{10}$ or PN respectively,
although inclusion of measurements of total $PM_{10}$ or PN can used instead, if available.

In the ideal case, if the individually computed factors for both data sets result in G(X) and
G(Z) being identical, then a straightforward joint model [X,Z] is successful and G[X,Z] = G(X)
= G(Z). However, if G(X) and G(Z) are significantly different then the joint model will fail,
identified by a too large Q value. A solution to this problem is to set the total weights of the
better dataset X significantly higher than the total weights of the auxiliary data set Z such
that X will "drive the model" and G[X,Z] will be approximately equal to G(X) and a reasonable
Q value is obtained for the Z. However, care is required to ensure that X or Z do not contain
rotational ambiguity because such rotation for X may not be suitable for Z. For such cases,
equal total weights for both X and Z are applied in the hope that the best rotation for both X
and Z can be found.

**2.2.4**    *2-Step method using data with different units - heterogeneous units*
The method proposed in this work separates the analysis of the two data sets X and Z into
two different PMF analyses. Dataset X is first analysed and an unambiguous rotation is
selected which gives computed factors G(X). These are then carried over into a second
PMF step in which G(X) are combined with Z to form a joint matrix for analysis. By using
FKEY (described below) factors, G(X,Z) are forced to be equal to G(X) from step 1. So for
example, if in the first step we analyse $PM_{10}$ data and carry forward the output $G(PM_{10})$ into
a second step combined with the NSD data, i.e. $[G(PM_{10}),NSD]$ this results in profiles
$F[G(PM_{10}),NSD]$. In other words, we force out of the NSD data source profiles which have
the same G factors as the $PM_{10}$ data and extend the list of components of the sources
identified in the first step and thus improve characterisation of the source. Note that this is
equivalent to non-negative weighted regression of matrix Z by columns of matrix G for which
other tools exist. Furthermore, by using a two step method, we can continue to use the
scaling method described in Section 2.2.2 to apportion the sources using a quantified time
series G(X) rather than normalising the G(X,Z) matrix sums to 1 and relying on the
summation of the elements in the rows of F(X,Z) to give the apportionment of X and Z. **2.2.5.**

***Application of PMF***
Positive Matrix Factorization was carried out in this work using the DOS based executable
file PMF2 v4.2 compiled by Pentti Paatero and released on Feb 11, 2010 (downloaded from
[www.helsinki.fi/~paatero/PMF/](www.helsinki.fi/~paatero/PMF/)).  This is used by the author in preference to a GUI version of
PMF (e.g. US EPA PMF 5.0, Norris et al., 2014) because of the ease with with it can be
incorporated into a Cran R procedure script using shell commands, thus facilitating
automation of the analysis and any optimisation.  R-script can be written to manipulate and
organise input data for PMF2, run PMF2, collect the output and produce the necessary
output for consideration as text, table or plot.  The main strength for this approach is to
improve the repeatability and transference of a method between practitioners within our
group.

The two step method is shown schematically in Figure 2.  Matrix X yields factors $^1G$ and $^1F$
in the first step.  The timeseries $^1G$ matrix is carried through to the second step where it is
combined with an auxiliary data set Z, to give the a step 2 input matrix [$^1G$ Z].  This in turn
is analysed to produce factors $^2G$  and $^2F$.  In the current example, the dataset of Beddows
et al.  (2015) is used as a starting matrix X and comprises the $PM_{10}$ chemical composition
dataset.  This yields timeseries $^1G$ and source profile $^1F$ and the reader is referred to
Beddows et al. (2015) for a description of the analysis and output.  Figure 1 shows the output
from the first step which was found to be the optimum solution after considering 3 to 8 factor
solutions.   The normalised timeseries matrix $^1G$ from this analysis was combined with the
NSD data - concurrently measured with the $PM_{10}$ data - to form the input matrix [$^1$GZ], for
step 2. The uncertainties of the $^1$G1 matrix, $^1\triangle$G are transferred from the output of the first
step and entered as input uncertainties for the second step.  The hourly NSD data was
aggregated into daily values to match the daily $^1$G factors outputted from the PMF analysis
of the daily $PM_{10}$ data sampled.  This reduced the data matrix down to 590 rows by 57
columns ($^1$G1…$^1$G6, $NSD_1^{16nm}$…$NSD_{51}^{640nm}$) for which a we have a $Q_{theory}$ value of 29748
for a 6 factor solution.  For the NSD data, the uncertainties are taken as the NSD values
multiplied by the value of an arbitrary parameter $b_{scale}$ (see Figure 2).  Initially, $b_{scale}$ was set
to 4 to to ensure that the model was weighted such that it was driven by the $PM_{10}$ data.
However, this operation becomes somewhat redundant by the use of the FKEY matrix
discussed in the next section.  However, in order to find the optimal NSD uncertainties the
value of the parameter $b_{scale}$ (typically, 0.2) was optimised in Cran R so that the ratio of
$Q/Q_{theory} = 1 \pm 0.02$, indicating an relative percentage uncertainty in the region of 20%.  In
retrospect – by taking into account the decrease in reliability of the size bin counts towards
the edges of the size bin range - an improvement would be to gradually increase the
uncertainties from 5% in the middle range of sizes to a pre defined larger value, e.g. 50%,
over the lower and upper size bins. The uncertainties were entered directly into the model
using PMF matrix T with U and V redundant.

**2.2.6   Pulling down with GKEY and FKEY**
GKEY and FKEY are matrices with the same dimensions as G and F respectively,  for
incorporating *a priori* information into a PMF analysis.  They are used in the second step of
the PMF analysis to "pull" elements of the source profiles to zero.  GKEY and FKEY indicate
the location of suspected zeros in source profiles $^2$F or contributions $^2$G (Figure S1). Since
we are concerned with the profiles, this information is given in the form of integer values in
an FKEY. The greater the certainty that an element of a source profile is zero, the larger the
integer value that is specified.  In this case, in the second step for the input dataset [$^1$G
NSD], it is certain that only one unique contribution will be strong for each row of the profile
$^2$F, outputted from the second PMF analysis, e.g. only $^1$G1 and not $^1$G2.. $^1$G6 will contribute
the to ($^1$G1, $^2$F$_1$) position in output factor $^2$F$_1$.  (Figure S1).  All 'non-zero' elements within
the output of $^2$F take a FKEY value of zero whereas all elements of $^2$F which are pulled to
zero take an non-zero value of $fkey_1$. This leads to a FKEY matrix which can be understood
in two parts.  The first part is a square matrix of dimension equal to the number of columns
of $^1$G with all its entries equal to $fkey_1$ except for the leading diagonal; this part ensures that
$^1$G is the same as $^2$G.  The second part of the matrix consist of all the elements as zero and
represents the NSD input data.  An $fkey_1$ value of 7 to 9 is considered a medium to strong
pull, and in this work, we used a value of 24 which in comparison is very aggressive ensuring
only one rotational solution is available ensuring $^1$G $\approx$ $^2$G.

To extend the analysis from 6 factors to 7 factors an extra row was added to FKEY.  This
was done in order to investigate any factors missed in the NSD data which the first analysis
using PM$_{10}$ would not be sensitive to.  For example, a nucleation mode would be detected
in NSD data but not PM$_{10}$ data.  In order to give the model freedom to factorise out a
nucleation factor, the 7$^{th}$ row of of FKEY values consisted {$fkey_1$, $fkey_2$… $fkey_6$,$nsd_1$, $nsd_{2…}$
$nsd_{51}$}.  This ensured that all the $^2$G contributions were allocted to the first 6 factors only
leaving the 7$^{th}$ factor to account for the remaining unfactorised NSD data.  There is no reason
why more than 7 factors could not be used to investigate possible unresolved NSD factors.
However, we constrained the scope of our investigation to reidentifying those in Figure 1.


### 2.3  Regression

As an alternative to using PMF in the second step, a regression was carried out. Each column of data for each of the 51 size bins $j$ within the NSD was regressed against the six $^1G$ time series using Equation 4

$$NSD_j = \alpha_{0,j} + \alpha_{1,j}\ ^1G_1 + \alpha_{2,j}\ ^1G_2 + \cdots + \alpha_{6,j}\ ^1G_6 \qquad (4)$$

where $\alpha_0$ is the population intercept and $\alpha_{1\text{-}6}$ are the populations slope coefficents. This results in a 7 by 51 matrix of values. Each column represents a size bin of the NSD data and each row represents the slope coefficients associated with 6 of the factors (giving an indication of how each size bin scales with each of the 6 factors) and an intercept. When $\alpha_{1\text{-}6,j}$ is plotted against the size bin, 6 plots showing the dependence of each size bin $j$ on each of the 6 $PM_{10}$ factors are produced. It is also assumed that these (referred to here as NSD regression source profiles) will be comparable to the actual NSD PMF source profile. Similarly, the $\alpha_{0,j}$ values are expected to give a background value due possibly to noise; however, it is more likely to yield a source (such nucleation) to which the $PM_{10}$ mass analysis is insensitive.

### 2.4  Peak Fitting

If it is assumed that the factors derived from the daily NSD data are the same as those present in the hourly data, i.e. the factors are conserved when averaging the data from hourly to daily data before PMF analysis, then daily NSD profiles can be fitted to the hourly NSD spectra to recover a diurnal cycle for the factors. However, it is worth noting that the process of aggregating hourly data to daily NSD data may cause loss of information implying that minor factors (e.g. due to event episodes) might well be averaged out of the data.

Given the $j^{th}$ size bin in the $i_{th}$ number size distribution $NSD_{i,j}$ (of dimensions M x N), the
factors can be fitted using equation (5).

$$D_i = \sum_{i=1}^{M} d_i \tag{5}$$

which is the $i_{th}$ sum $D_i$ of the difference ($d_i$ give by equation 6) across the size bins of the $i_{th}$
$NSD_i$ and the linear sum of the $p$ NSD source profiles ($p = 7$ in this case) scaled with respect
to the scalar values $c_{ik}$, representing the timeseries of each fitted NSD source profile.

$$d_i = \sum_{j=1}^{N} \left\{ NSD_{ij} - \sum_{k=0}^{p} c_{ik} \times f_{kj} \right\}, \qquad c_{ik} \geq 0 \tag{6}$$
$$1 \times 10^{10}, \qquad c_{ik} < 0$$


The Cran R package Non-Linear Minimization (nlm) (R Core Team, 2018) was used to
minimise the value of $D_i$ with respect to the scalar values $c_{ik}$ with a non-negative constraint
on $c_{ik}$ placed in the function. If a negative value is returned by any of the $c_k$ values then $D$
returns an excessively large value. Furthermore, in order to extract an apportionment to
number concentration ($1/cm^3$) the fitted values were scaled using a scalar $\beta_k$. Seven values
were derived for $\beta_k$ by regressing the total particle number (total hourly SMPS) against each
of the fitted values $c_k$ (equation 7.

$$PN = \beta_0 + \beta_1 c_1 + \beta_2 c_2 + \cdots + \beta_7 c_7 \tag{7}$$

The resulting scaled-fitted values were then used to calculate the PN concentration for each
of the regression source profiles (equation 8) allowing subsequent plotting of the 7 diurnal
cycles.

$$PN_k = \beta_k c_k \tag{8}$$



## 2.5    Bivariate Plot

Identification of the sources responsible for the factors outputted from PMF can be assisted by meteorological data. Time series of the $k_{th}$ factor (or $g_k$ values) can be plotted against wind direction and wind speed using either the polarPlot or polarAnnulus functions provided in the Openair package. Polar Plots are simply used for plotting the factor contribution on a polar coordinate plot with North, East, South and West axes. Mean concentrations are calculated for wind speed-direction 'bins' (e.g. 0-1, 1-2 m/s,... and 0-10, 10-20 degrees etc.) and smoothed using a generalized additive model. Each bin concentration is plotted as a group of pixels (coloured according to a concentration-colour scale) and positioned a distance away from the origin according to the magnitude of wind speed and along an angle from the North axis according to the wind direction. Such plots are useful when identifying the nature of the source. A diffuse source will tend to have its highest concentration showing as a *hotspot* at the origin of the polar plot, whereas a point source will cause a *hotspot* both away from the origin and in the direction pointing towards the source. On the other hand wind blown sources tend to be recognised by their relation to wind speed and hence do not necessarily produce *hotspots*. Instead, they produce a minimum to maximum gradual gradient of colour from the origin, spreading radially out towards the edge of the plot in the direction of the source, e.g. for a marine source. Likewise, Annulus Plots plot the mean factor concentration on a colour scale by wind direction and as a function of hour-of-the-day as an annulus, represented by the distance of the coloured pixels from the origin. The function is good for visualising how concentrations of pollutants vary by wind direction and hour of the day. For example, for the North Kensington site – positioned West of the city centre – we might well expect most of the anthropogenic sources (traffic, diffuse urban, etc) to show an Easterly direction with the appropriate diurnal cycle (e.g. rush hour traffic patterns). Similarly, we might expect cleaner air (Marine, Nucleation, etc) to occur from a

Westerly direction and at times of the day when the solar strength is highest.

**3.      RESULTS AND DISCUSSION**
The aim of this work has been to show how a given PMF result can be complemented with
concurrently measured auxillary data. We exemplify this using $PM_{10}$ and NSD data collected
from the North Kensington receptor site in London and start with the premise that we are
completely satisfied with the $PM_{10}$ analysis and are using a rotation which gives quantified
factors (quantified G and scaled F) which best represent the urban atmosphere sampled,
i.e. the output from Beddows et al. (2015).   For each $PM_{10}$ factor we wish to assign a NSD
distribution. Rather than repeat the PMF analysis using a combined $PM_{10}$+NSD dataset
which can be complicated if the rotations of the individual PMF analyses of $PM_{10}$ and NSD
data are mismatched or ambiguous, we can carry out a  a second PMF analysis or a
regression.

Furthermore, by the nature of any factor analysis, we also have to make the assumption that
each source chemical profile and size distribution not only remain unchanged between
source and receptor but that it remains constant throughout the measurement campaign.
This of course limits our capacity to fully understand the aerosol within the atmosphere we
are considering. Chemical reactions during the transit of the air masses will of course modify
the chemical composition. It might be assumed that a fully aged aerosol remains unchanged
and is identified as a background component, but for example we would expect progressive
chlorine depletion within a fresh marine aerosol passing over a city. Likewise, we also have
to appreciate that different particle sizes will have different atmospheric transit efficiencies
with large particles settling out of the air mass before smaller ones. Similarly, particles
nucleate and grow from 1 nm up to 20-30 nm over a short time period of time. It is these
finer details which are missed when making an overall assessment of the chemical and
physical composition of an air mass measured over a long period (e.g. 2 years) dataset
using PMF.

**3.1    2-Step PMF-PMF Analysis**
Figure 3 presents   the profiles $^1F_k$ and $^2F_k$   from the first and second PMF analysis
respectively.  The plots of $^1F_k$ were carried over from Beddows et al. (2015) to complete the
assignment of the source profiles.

The  time series $^1G_k$ and uncertainties $^1\triangle G_k$ from the first PMF analysis of $PM_{10}$ data were
carried over into the second step where they are combined with the NSD data for PMF
analysis (Figure 2).  The uncertainties of the NSD data are taken as an optimised multiple
of the NSD values themselves (~ 5 % uncertainty, yielding a Q value of 30,333 in the robust
mode; see Table S2 for PMF settings).  Also in order to  encourage $^2G_k$ to be proportional
to $^1G_k$ for $k$ = 1-6 (see Table S4), the FKEY matrix is applied to pull elements in the source
matrix to zero as described in section 2.3.3. This ensured that the PMF analysis of the NSD
data was driven by the $^1G$ time series and  resulted in a 6 factor output in which there were
unique contributions from the $k_{th}$ factor $^1G_k$ from the first analysis to the $k_{th}$ factor $^2F_k$  in the
second analysis.   This is mainly due to the aggressive pulling of the factor element in $^2F$
applied using FKEY.

When inspecting Figure 3 it is notable that the source profiles are surprisingly similar to
those calculated for the just-NSD and $PM_{10}$+NSD data in Beddows et al. (2015). The Diffuse
Urban factor has a modal-diameter just below 0.1 $\mu$m which is comparable to the same

factor in the just-NSD analysis.  Marine is comparable to the Aged Marine factor derived

from the $PM_{10}$+NSD analysis. The Secondary factor is again the factor with the largest modal

diameter (between 0.4 and 0.5 $\mu$m) and traffic has as expected a modal diameter between

30 and 40 nm.  The Fuel Oil factor appears to be a combination of a nucleation factor and a

mode comparable to diesel exhaust seen in the Traffic factor.

**3.2     2-Step PMF-LR Analysis**

Figure S2 shows the results of the linear regression of the NSD data plotted against the

$PM_{10}$ $^1G_k$ scores and again what is remarkable is the similarity between these regression

source profiles and both the factors derived in Beddows et al. (2015) and those from the 2-

step PMF-PMF analysis.

This PMF-LR analysis was carried out using daily averaged data and to obtain hourly

information - and thus obtain the diurnal patterns (Figure S2) - the resulting regression

source profiles were re-fitted to the original NSD data.  On inspection of these source profiles

and diurnal plots, the negative values make interpretation a struggle reinforcing one of the

4 conditions (Hopke, 1991) in the analysis if it is to make sense. We can however fit non-

negative gradients using non-negative regression.  However, the surprising consequence of

applying this constraint is that the same profiles are derived but they are clipped so that all

negative values are replaced by zero values – hence, information is lost by doing this.   One

interpretation of the negative values is that these are particle sinks but this contradicts the

PMF-PMF findings and hence it is concluded that the PMF-LR analysis only serves as an

indication of how the $PM_{10}$ factors are augmented by the NSD data.  If all profiles are shifted

to above the zero line then comparisons to the PMF-PMF data can be made.   However,

what is interesting to note in this result is the intercept NSD which is comparable in profile
and diurnal pattern to the nucleation mode identified in Beddows et al. (2015).  This is a
seventh regression source profile, in addition to the 6 $PM_{10}$ factors and suggests that
although the PMF analysis of the $PM_{10}$ data alone misses a Nucleation factor, this can be
recovered in a second analysis as a remainder or bias in the data.  Furthermore, this result
indicates that the composition of the Nucleation NSD factor has no link to the chemical $PM_{10}$
composition and cannot be used to infer a composition. This is unsurprising given the very
small mass contributed by the nucleation mode particles.

Returning to the PMF-PMF analysis and extending the analysis from 6 factors to 7 factors,
an extra row in the FKEY  matrix was added to pull all of the $^1G_7$ contributions to $^2F_7$ to zero
in the solution (Figure S1).  The same FKEY matrix of $fkey_1$ and 0 values was used but this
time it was augmented with a $7^{th}$ row of $fkey_2$ and zero values.  In this case, the $fkey_2$ values
were set to a value of 20.

The same 6 factor solution is obtained with the additional $7^{th}$ factor (Figure 4 and Figure S3)
and as expected, this seventh factor was a Nucleation factor.  It was suspected that in the
6 factor solution, the Nucleation factor was combined with the Fuel-Oil factor.  This does not
suggest any link between the Nucleation and Fuel-Oil factor other than there was an
insufficient number of factors within the model for the two to factorise out of the data giving
the Fuel-Oil NSD profile a more reasonable modal peak between 50 and 60 nm rather than
20, 30 and 60 nm.

Beddows et al. (2015), applied a 1-step analysis to three different datasets: $PM_{10}$-only; NSD-

only and PM$_{10}$+NSD. The analyses of the PM$_{10}$-only and NSD-only – both with homogeneous units - produced quantitative timeseries G. This was unlike the analysis of the PM$_{10}$+NSD with heterogeneous units which could not apportion its 5 factors using G but was able to factorise out a Nucleation factor from the data, seen also in the 4 sources in the PMF solution for the NSD-only data. A PM$_{10}$-only seven factor solution did not reveal this factor, presumably because the mass associated with nucleation mode particles is too small to affect composition significantly. Furthermore, Fuel Oil was not factorised out of the PM$_{10}$+NSD data and was more likely divided across all 5 factors.

Another interesting observation is that although only 4 factors were derived from the PMF analysis of NSD-alone (Diffuse Urban; Secondary; Traffic and Nucleation), when extra information is included from the PMF analysis of the PM$_{10}$ data, more information can be extracted from the PMF analysis of the NSD data in the form of the Marine; Fuel Oil and NET & Crustal factors. The Nucleation factor is only revealed when performing a regression between the NSD size bins and the G scores of the PM$_{10}$ PMF analysis which leads to increasing the factor number from 6 to 7 which yields the Nucleation profile. It is also reassuring that the bivariate plots for the 7 factors (discussed in the next section) correspond to the bivariate plots given in Beddows et al. (2015). Also note that there is no reason why any further investigation might not explore using more than 7 factors. In fact the Nucleation factor appears at first sight to be multimodal. However, we restricted our analysis to 7 factors, considering it complete in terms of identifying the sources obtained by Beddows et al. (2015).

**3.3      Diurnal and Bivariate Plots**

The original PMF was carried out on daily $PM_{10}$ data and in order to make diurnal and bivariate plots, a higher time resolution is desirable.  It is assumed that the factors derived in the hourly NSD data are the same as those derived from the daily averaged data, i.e. the factors are conserved when averaging the data from hourly to daily data before PMF analysis.  Then the hourly NSD data can be fit with the PMF profiles derived from the daily data (see Section 2.4).  Figure 5 shows the resulting diurnal profiles.  The diurnal trends of the parameter $c_k$ (equation 7), required to fit the 7 daily NSD factors to the hourly NSD data are shown.  These have been scaled to PN (measured in $1/cm^3$) using the integral of the NSD (equation 8).  The Nucleation factor diurnal trend behaves as expected rising to a maximum during the day and then falling back down to a minimum at night.  This corresponds to the intensity of the sun during the day and the increased likelihood of nucleation on clean days when there is sufficient precursor material to form particles with a low particle condensation sink.  The Marine factor is also high during the day presumably due to higher wind speeds. Diffuse Urban, NET & Crustal, and Traffic all follow a trend which is synchronised to the daily cycle of anthropogenic activity and traffic as influenced by greater atmospheric stability at night.  The Secondary factor shows a small diurnal range. Fuel Oil is highest during the evening and night and may correspond to home heating rather than shipping emissions. The particle size distributions associated with the Marine and NET & Crustal sources are of limited value as these sources are dominated by coarse particles, beyond the range of the SMPS data, although there is a sharp increase in the volume of the particles above 0.5 $\mu$m in the Marine factor.  As pointed out in Beddows et al. (2015), the Marine factor is identified by its chemical profile of sodium and chloride and is accompanied by an aged nucleation mode at around 30nm.  This can be either viewed simply as clean marine air being 'polluted' by traffic emission and/or as the consequence of nucleation

occuring over at city in clean maritime air masses (Brines et al. 2015). The key point here
is that the factors derived in this work are comparable to those factorised in Beddows et al.
(2015) using the combined dataset and the advantage of the 2-step approach is that now
we have quantified hourly timeseries G.

The hourly contributions are aggregated into daily values and plotted as bivariate plots in
Figure 5 to assist comparison with the daily plots in Beddows et al. (2015). In that work, the
same PMF analysis of the NSD data yielded 4 factors which are named identically to those
in the bivariate plots. The similarity of both of the polar and annular plots for each of the 4
factors supports our previous factor identification. The Secondary and Diffuse Urban are
background sources with strongest contributions in the evening and morning. Traffic is
strongest for all wind speeds from the East which makes sense since North Kensington is
to the West of the city centre of London where traffic is expecting to be most dense.
Nucleation is also seen to be strongest for those wind direction from the West which are
expected to be cleaner, and have a lower condensation sink. NET & Crustal and Fuel Oil
are similar to Diffuse Urban suggesting a similar predominant source location in the centre
of London. Marine is observed to be strongest for elevated wind speeds for all wind
directions which is consistent with the expected strong contribution for all high wind speeds
from the South West, as observed in the daily polar plots in Beddows et al. (2015).

**3.4     Composition associated with the Nucleation Factor**
The Nucleation factor was extracted from the two-step PMF-PMF analysis which included
pulling the $^{1}G_{1}$-$^{1}G_{6}$ to zero of factor $^{2}F_{7}$. It might be reasonable to suggest that if the two-
step PMF-PMF analysis is repeated and the order of analysis of $PM_{10}$ and NSD datasets

reversed that it would be possible to derive the chemical conditions within the atmosphere which were conducive to nucleation. For this, the time series of the 4 NSD factors ($^1G_1$-$^1G_4$) reported in Beddows et al. (2015) were combined with the PM$_{10}$ data. We again assume that the first PMF step has been carried out and that we are satisified with how the final solution represents the urban environment of the receptor site and that there are no rotational ambiguities. We then carry out the second step PMF analysis on the 34 x 591 input matrix ($[^1G1…^1G4]$, PM$_{10}$[PM,PM$_{carbon}$,PM$_{ions}$,PM$_{metals}$]). The hourly output uncertainies from the first PMF analysis of the NSD data $^1\Delta G1…^1\Delta G4$ were carried forward into the second PMF analysis by adding them *in quadrature* to give daily uncertainties. As with the analysis of the auxillary data in the PM$_{10}$-NSD data, the measurement uncertainties for the PM$_{10}$ data (this time the auxillary data) was naively taken as 4 times the PM$_{10}$ matrix. Extra care could have been take in assigning the PM$_{10}$ uncertainties but since we force the output using FKEY a simpler approach was taken. In fact, the FKEY consisted of a 4 x 4 diagonal matrix of zero values with an *fkey*$_1$ of 20 for all the off-diagonal positions joined to a 4 x 30 matrix of zeros. Furthermore, the uncertainty values of the PM$_{10}$ were scaled until Q/Q$_{theory}$ = 0.99 using parameter $b_{scale}$ = 0.35 (see Table S3 for more details).

Ideally, the chemical data would be limited to the composition of the particles in the same size range as the SMPS data. However, when since we are using the PM$_{10}$ composition data we can at best describe the composition of the aerosol which accompanied each factor (Figure S4). For the NSD Secondary factor with its strongest contribution (indicated by the Explained Variation) ~400 nm, we have a strong contribution to PM$_{10}$ and PM$_{2.5}$ together with nitrate, sulphate and ammonium. Diffuse Urban, with its strongest contribution at 100 nm is accompanied by contributions from elemental carbon and wood smoke indicative of traffic and recreational wood burning. There are also contributions from barium, chromium,

iron, molybdenum, antimony and vanadium, all indicative of non-exhaust traffic emissions
and the burning of fuel oil.  Similarly, the Traffic factor has a modal diameter at roughly 30
nm which is indicative of exhaust emissions and this is accompanied by contributions to
aluminum, barium, calcium, copper, iron, manganese, titanium and various other metals
attributed to vehicles, albeit from tyre or brake wear or resuspension.

The Nucleation factor with its peak ~20 nm, was associated with marine air as indicated by
the strong contributions to Na, Cl and Mg (Figure S4).  There are also traces of V, Cr, Ni
and a high contribution to $PM_{10}$ mass which are all associated with marine air.  This is
explained by an association with the south-westerly wind sector which brings strong winds
and marine aerosol rather than reflecting the composition of the nucleation particles
themselves.  Marine air is considered to provide the conditions required of an air mass
conducive to nucleation, i.e. cleaner air with particles with a low condensation sink.  As these
air masses pass over the land and eventually into London, anthropogenic precursor gases
are added to this air which then nucleate particles seen at the receptor site as a nucleation
mode.  This also goes some way to explain the earlier observation of aged nucleation
particles observed in the marine factor in Figure S3.  There are also strong contributions to
vanadium which is most likely from an unresolved Fuel Oil source being mixed into the
Marine and Diffuse Urban factors.

**4.    CONCLUSIONS**
A two-step PMF analysis method is presented whereby existing PMF profiles can be extend
to incorporate auxillary data concurrently measured and having different units.  This is
exemplified using $PM_{10}$ and NSD data.

When analysing $PM_{10}$ data, the inclusion of auxillary data such as meteorological, gas and
particle number data has proved to give a clearer separation of factors. However, for a
successful output, there must be no rotational ambiguity in either the $PM_{10}$ data or in the
auxillary data. In the ideal case, the individually computed factors G(X), G(Z) and G(X,Z)
need to be similar if the joint model is to be successful and not produce large residuals and
hence a too large Q value. In the best case, the total weight of the $PM_{10}$ data can be set
higher than the auxillary data so that the $PM_{10}$ data drives the analysis. In this work, we
present an alternative method called the 2-step PMF method. In the first step the $PM_{10}$ data
is PMF analysed using the standard approach without the inclusion of additional data. An
appropriate solution is derived using the methods described in the literature in order to give
an initial separation of source factors. The time series G (and errors) of the $PM_{10}$ solution
are then taken forward into the second step where they are combined with the NSD data.
The PMF analysis is then repeated using the combined and mixed unit G time series and
NSD dataset. In order to ensure that unique factors are obtained for the G scores, FKEY is
used to pull off-diagonal values to zero thus driving the NSD data. This ensures that the
NSD factors are specific to the $PM_{10}$ solution and the $PM_{10}$ analysis is not affected by any
rotational ambiguity of the NSD data. For our demonstration using the Beddows et al. (2015)
analysis, this results in 6 $PM_{10}$ factors whose time series are not only apportioned in mass
but the source profiles are identified for the NSD data. Comparisons of both the factor
profiles, diurnal trends and bivariate plots to those of Beddows et al. (2015), show that this
technique produces one solution linking the two separate solutions for $PM_{10}$ and NSD data
datasets together. This generates confidence that the NSD and $PM_{10}$ factors ascribed to
one source are in fact attributable to that same source.

Hence, the process starts with a dataset which produces a solution which is sensitive to
mass but the factors more sensitive to number can be accessed using a second step.
Furthermore, by exploring a higher number of factors, NSD factors which are insensitive to
$PM_{10}$ mass can be identified as in the case of the Nucleation factor. This information can
also be extracted using a linear regression PMF-LR where the size bins of the NSD data are
regressed against the $PM_{10}$ PMF time series. For this dataset, the Nucleation factor profile
is identified as an intercept within the fitted model leading to an increase in the number of
PMF factors from 6 to 7.

**5.        ACKNOWLEDGEMENTS**
The National Centre for Atmospheric Science is funded by the U.K. Natural Environment
Research Council. Figures were produced using CRAN R and Openair (R Core Team, 2016;
Carslaw and Ropkins, 2012).

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

**FIGURE LEGENDS:**

**Figure 1.** Venn Diagram showing the summary of the findings of Beddows et al. (2015); applying PMF to $PM_{10}$-only, NSD-only and $PM_{10}$+NSD datasets. Table shows the apportionment of $PM_{10}$ and NSD taken from Beddows et al. (2015).

**Figure 2.** Flow diagram showing the flow of data through the 2-step PMF-PMF analysis. The PMF analyses of single data set X are considered in step 1 and output indicated by factors/uncertainties $^1G$, $^1\triangle G$, $^1F$ and $^1\triangle F$. The second PMF analysis is carried out on the joint data set [$^1GZ$] and yields factors/uncertainties i$^2G$, $^2\triangle G$, $^2F$ and $^2\triangle F$. In our analysis, X and $^1G$ are the $PM_{10}$ and resulting time series from the analysis of Beddows et al. (2015) and Z is the auxillary NSD data concurrently measured using a SMPS.

**Figure 3.** Source profiles $^1F$ and $^2F$ from both the first and second PMF step using 6 factors. [Grey bars and black line indicate the values of F; red lines and dots indicate the explained variations; and grey dotted line indicates the dV/dlogDp.].

**Figure 4.** Nucleation and Fuel Oil factors derived when extending the second PMF analysis from the 6 factors (shown in Figure 3) to 7 factors. Source profiles $^2F_1$ to $^2F_6$ are given in Figure S3. Each plot is divided into 2 showing the output $^1F_k$ and $^2F_k$. [Grey bars and black line indicate the values of F; red lines and dots indicate the explained variations; and grey dotted line indicates the dV/dlogDp.]

**Figure 5.** Diurnal cycles derived $PN_k$ calculated by the fitting of the daily PMF factor profiles to the hourly NSD data fitted (see equation 8 and Section 2.4). [Left-left column – diurnal trends of $PN_k$; left-middle column – bivariate plot of $PN_k$; middle-right – annular plot $PN_k$; right-right – bivariate plot of $PN_k$, plotted using the Openair program. Polar plots show a point coloured acording to the key, the number concentration at that point on the plot whose distance from the origin represents wind speed and angle wind direction. Likewise for the angular plots the number concentration represent wind direction at an hour of the day between 0 and 23 hrs.]. Note that the diurnal plots do not start at zero.



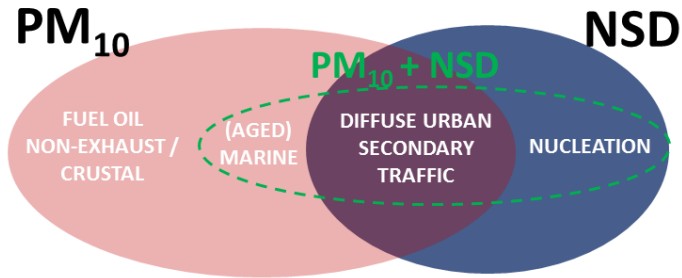

| | PM₁₀ [μg/m³] | NSD [1/cm³] |
|---|---|---|
| Diffuse Urban | 4.1 | 2370 |
| Marine | 2.6 | - |
| Secondary | 4.4 | 243 |
| NET / Crustal | 4.3 | - |
| Fuel Oil | 1.0 | - |
| Traffic | 0.8 | 2460 |
| Nucleation | - | 430 |
| Total | 17.2 | 5512 |

**Figure 1.** Venn Diagram showing the summary of the findings of Beddows et al. (2015); applying PMF to PM₁₀-only, NSD-only and PM₁₀+NSD datasets. Table shows the apportionment of PM₁₀ and NSD taken from Beddows et al. (2015).



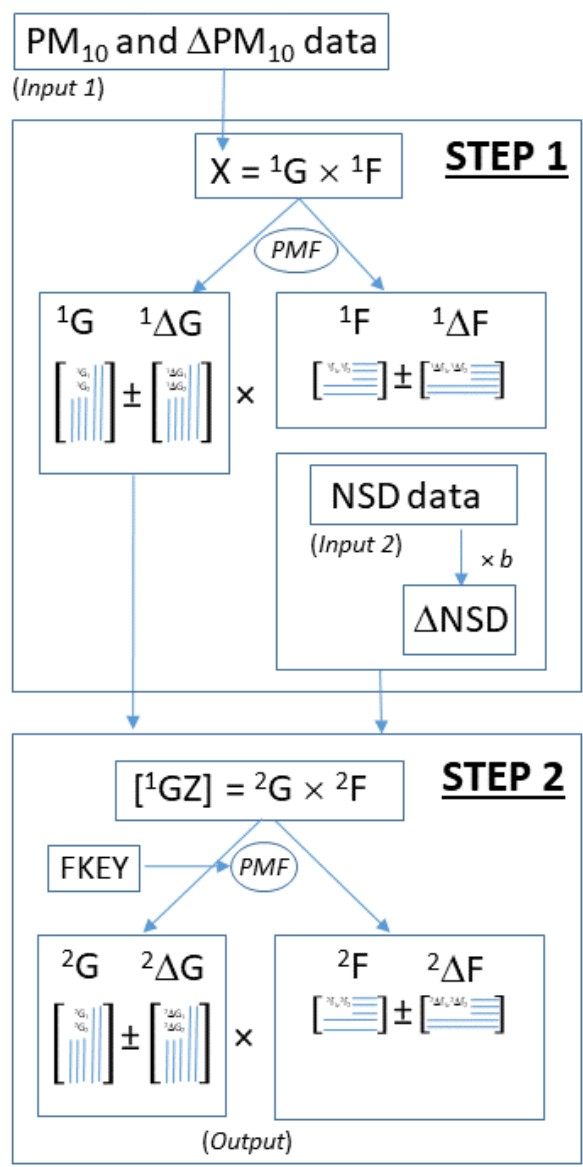

**Figure 2.** Flow diagram showing the flow of data through the 2-step PMF-PMF analysis. The PMF analyses of single data set X are considered in step 1 and output indicated by factors/uncertainties $^1G$, $^1\triangle G$, $^1F$ and $^1\triangle F$. The second PMF analysis is carried out on the joint data set $[^1GZ]$ and yields factors/uncertainties i$^2G$, $^2\triangle G$, $^2F$ and $^2\triangle F$. In our analysis, X and $^1G$ are the $PM_{10}$ and resulting time series from the analysis of Beddows et al. (2015) and Z is the auxillary NSD data concurrently measured using a SMPS.

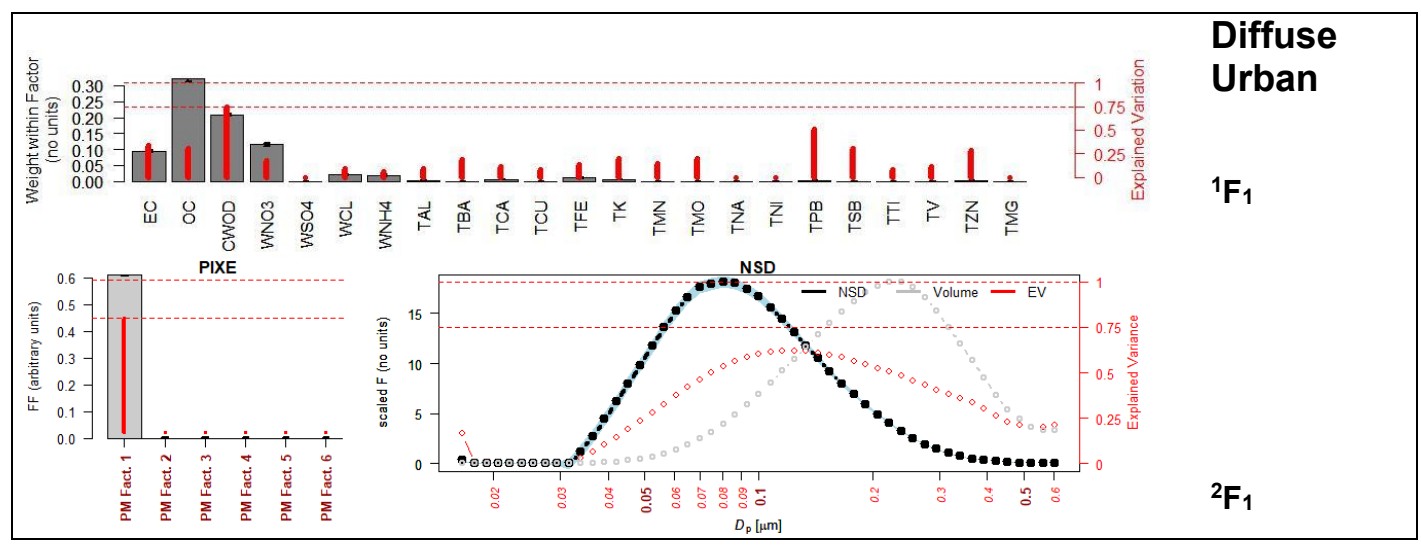

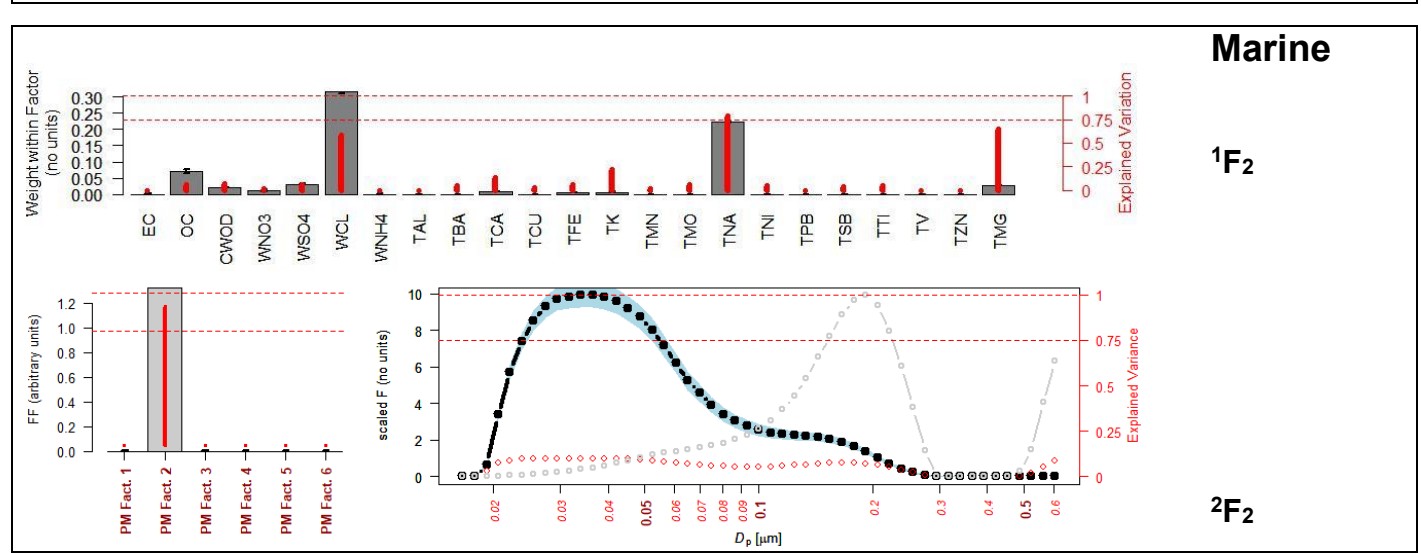

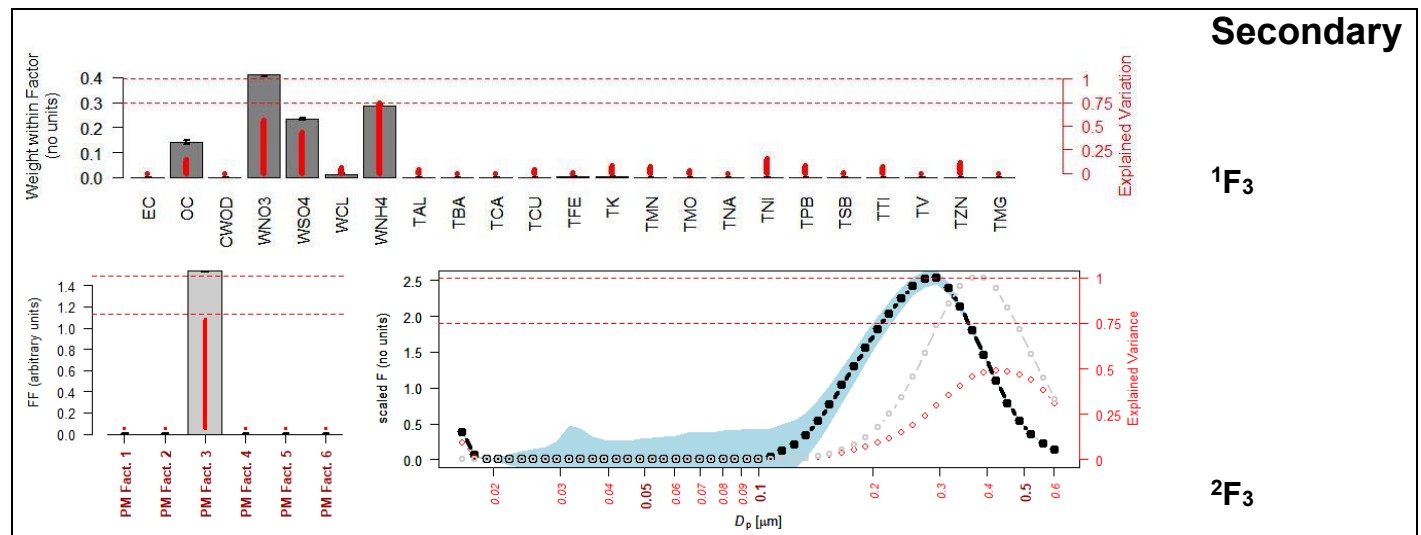

869

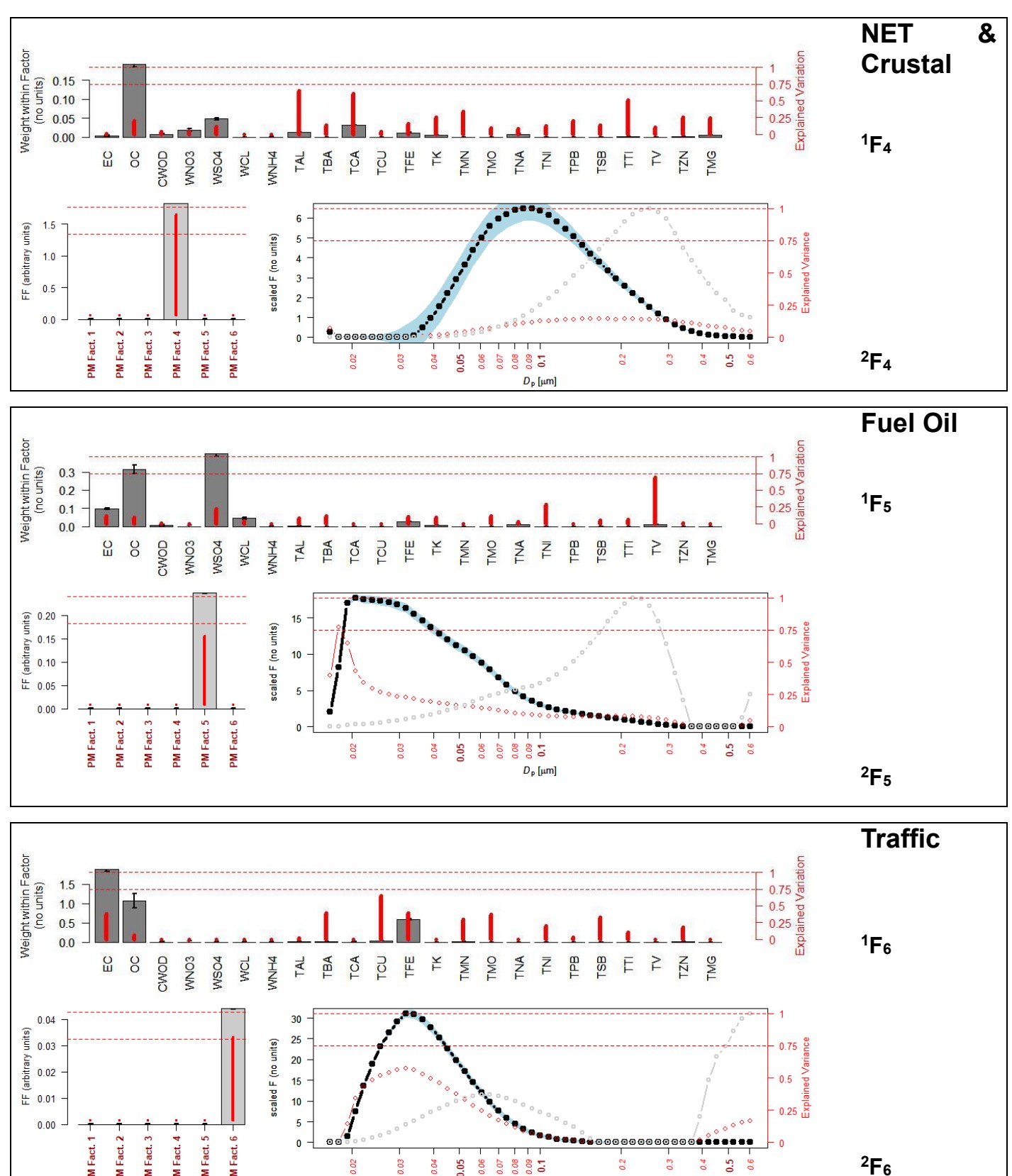

**Figure 3.** Source profiles [1]F and [2]F from both the first and second PMF step using 6 factors. [Grey bars and black line indicate the values of F; red lines and dots indicate the explained variations; and grey dotted line indicates the dV/dlogDp.]

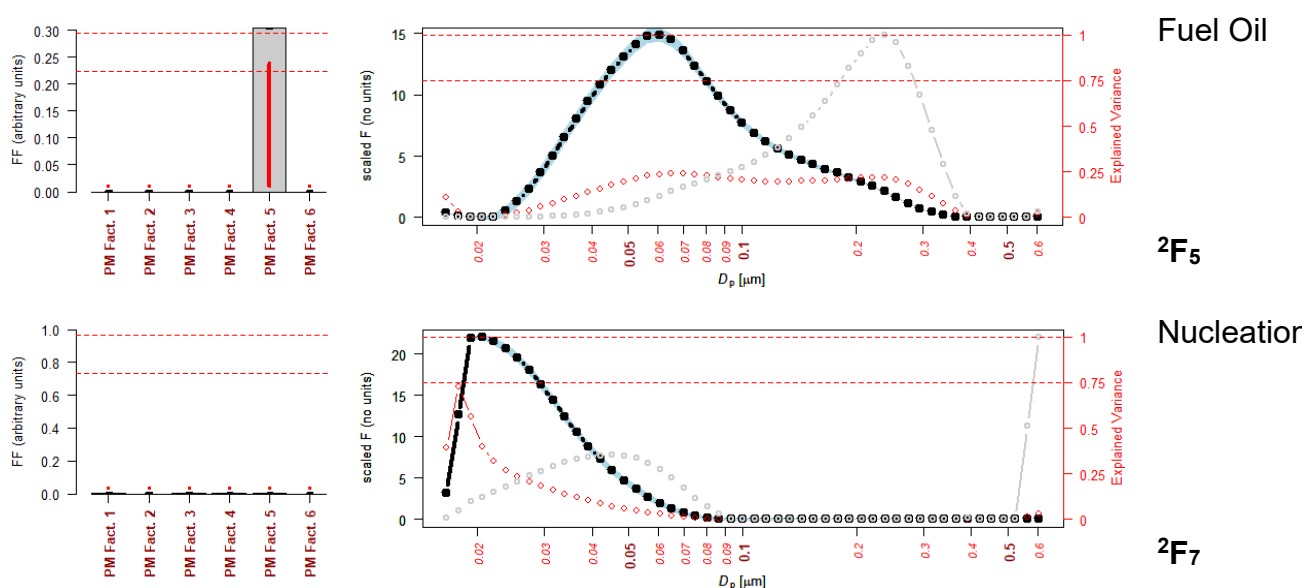

**Figure 4.** Nucleation and Fuel Oil factors derived when extending the second PMF analysis from the 6 factors (shown in Figure 3) to 7 factors. Source profiles $^2F_1$ to $^2F_6$ are given in Figure S3. Each plot is divided into 2 showing the output $^1F_k$ and $^2F_k$. [Grey bars and black line indicate the values of F; red lines and dots indicate the explained variations; and grey dotted line indicates the dV/dlogDp.]

880

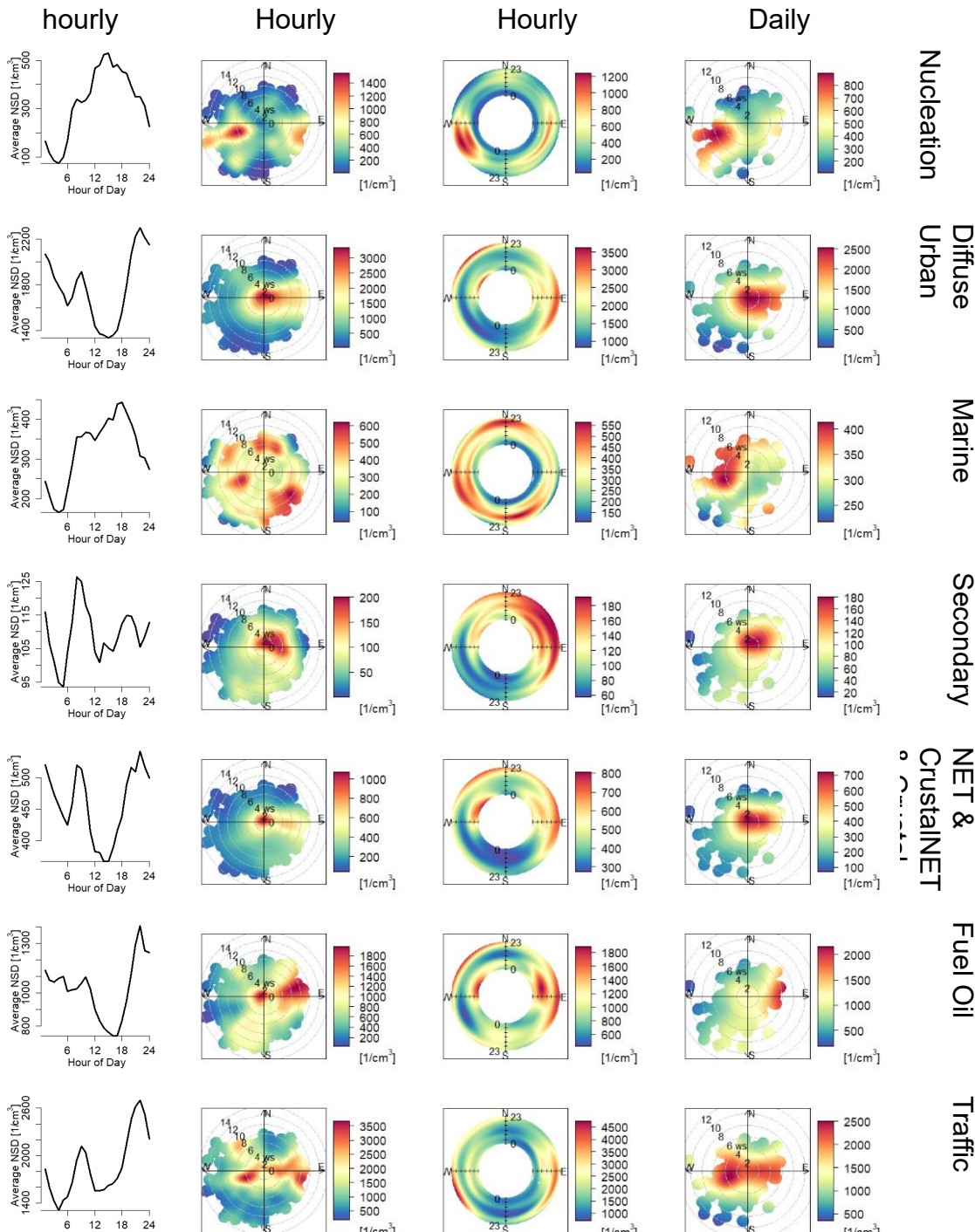

881

**Figure 5.** Diurnal cycles derived $PN_k$ calculated by the fitting of the daily PMF factor profiles to the hourly NSD data fitted (see equation 8 and Section 2.4). [Left-left column – diurnal trends of $PN_k$; left-middle column – bivariate plot of $PN_k$; middle-right – annular plot $PN_k$; right-right – bivariate plot of $PN_k$, plotted using the Openair program. Polar plots show a point coloured acording to the key, the number concentration at that point on the plot whose distance from the origin represents wind speed and angle wind direction. Likewise for the angular plots the number concentration represent wind direction at an hour of the day between 0 and 23 hrs.]. Note that the diurnal plots do not start at zero.

890