# Peer review of "RECEPTOR MODELLING OF BOTH PARTICLE COMPOSITION AND SIZE DISTRIBUTION FROM A BACKGROUND SITE IN LONDON, UK – THE TWO STEP APPROACH"

_Atmospheric Chemistry and Physics, 2018_

## Referee Comment (RC1) · Anonymous Referee #1 · 21 Sep 2018

General comment The paper regards the description of a two-step approach for performing source apportionment using the PMF receptor model and an input composed by variables having different measurement units. The approach has elements of originality and potentially several applications. The topic is interesting considering that source apportionment is a major topic in nowadays research and the possibility to use an approach that use input variables having heterogeneous measurement units is certainly appealing. I also believe that the topic is suitable for the Journal and the paper generally well written and understandable. However, I found that some aspects are

not completely clear (see my specific comments) and the paper need a revision before publication.

Specific comments

Lines 59-62. This sentence is not completely true and I would suggest to modify it. This happens only if number size distributions are mixed with chemical composition (in mass), however, there are examples in which size-segregated chemical composition is used in PMF analysis to obtain quantitative evaluation of size distribution of sources (see for example Contini et al., 2014 Science of the Total Environment 472, 248–261 and references therein).

Lines 140-141. Better to write 16-604 nm (like in line 149) because two decimal digits for size is an illusory precision.

Lines 159-161. The conversion of mass difference in PN0.6-10 is likely quite uncertain. Some details should be given because I believe that some assumptions have been done regarding size distribution in the range o.6-10 micron and the result of the conversion would be strongly influenced by these assumptions. A comment on this aspect is needed.

Lines 312-314. This could happen because nanoparticles have a limited mass to influence significantly PM10 mass composition, however, it could be different if NSD are mixed with PM1 chemical composition for example. A comment on this aspect would be useful.

There is a particular reason for using PMF2 and not the more advanced PMF5 that is becoming the standard version of source apportionment with PMF?

Lines 215-216. How much is it the X value chosen? This should likely be reported for completeness.

Line 259. I believe that the number of factors is six rather than seven.

Lines 358-360. Looking at figures 4 and S3, it seems that the marine source is dominated by nanoparticles. Considering that this is a source generally made of coarse particles, and also authors mention this aspect, this result appears unusual and some discussion and explanations are needed.

Minor comments

Lines 148. "spherical"

Line 351. "there is…"

The source "NET and crustal" is reported in the text but repeated in the figures as "NET and coarse". I would suggest to use "NET and crustal" in all the paper that is more understandable and appropriate.

Title Section 3.4. Why hidden? Moreover, this section s dedicated to several factors…what is the hidden one the nucleation? An explanation or a change of the title is needed.

What is the meaning of the "*" reported in figures 4 and 5?

---

## Referee Comment (RC2) · P. Paatero (Referee) · 5 Nov 2018

RECEPTOR MODELLING OF BOTH PARTICLE COMPOSITION AND SIZE DISTRI-BUTION FROM A BACKGROUND SITE IN LONDON, UK – THE TWO STEP AP-PROACH

by David C.S. Beddows and Roy M. Harrison

submitted to ACP

This manuscript deals with PMF analyses of "combined" data matrices such as [X Z] where X contains elemental composition profiles of aerosol samples and Z contains aerosol number size distributions measured simultaneously with composition profiles. This is an important problem that occurs often in modern aerosol research. There are specific problems in this task; these problems have not been studied in depth in literature so far.

This manuscript studies one specific combined data matrix and reports a PMF model for this matrix. Thus the ms might deserve publication despite of certain serious problems. These problems are in part related to misunderstandings found in earlier papers that discuss this same topic. For this reason, the present review contains a lengthy general discussion of the task of modeling combined matrices. The specific questions regarding this ms are based on this general discussion.

The ms might also be suitable for publication in the sister Journal AMT, Atmospheric Measurement Techniques. My personal view is slightly in favour of AMT. However, both ACP and AMT seem possible, and this review considers publication in either Journal.

The structure of this review is as follows:

=====================================================

Recommendations

Notation used in this review

Background

Common mode errors

Joint matrices containing different units

Discussion of the manuscript

Two-stage PMF model vs. customary PMF model

[Figure]

The hidden factor, aka Nucleation factor

Miscellaneous

==================================================

Recommendations

There are very many problems of different kinds in this manuscript. For this reason, I hesitantly recommend that this ms should NOT be published by ACP or AMT. However, if it is desired to publish this ms because of the importance of the problem, then a thorough rewriting of the text and mathematical details must be undertaken. I recommend that the following enhancements be performed:

There has apparently been lack of communication between the person(s) who did the actual computations and those who wrote the paper. For this reason, the mathematical description is erratic, chaotic and impossible to understand or replicate. In order to create an accurate description, the person(s) who did the computations should be included in the group of authors. Without such help, it may not be possible to achieve a satisfactory mathematical description of what was done.

The entire mathematical discussion about problems attributed to PMF analysis of joint matrices containing different units is erroneous, based on a widespread misunderstanding. This discussion must be rewritten according to suggestions given below. It might be good to include in the author group somebody familiar with the quantitative mathematical structure of the PMF model. In particular, it seems that lines 79,80 are not based on quantitative understanding of the model. These lines, and other similar sentences, must be removed.

Much of Conclusions must be rewritten so that the claims against using variables with different dimensions/units are replaced by opposite sentences stating e.g. that a joint analysis of matrices of variables with different dimensions/units is not harmed by these differences but unfortunately the opposite was believed to be true when the work was

carried out.

The mathematical description of what was done must be totally rewritten so that systematic matrix-form notation is used. Equations must be corrected and written in correct notation, using correct terminology and correct numbering. Details of PMF modeling must be reported, such as dimensions of matrices, used parameters such as uncertainties of data values, robust/nonrobust, obtained Q values, numbers of observed outliers, unique or multiple minima, and so on.

Rotational questions are an ever-present problem in factor analytic modeling, independently of what programs are used. It is alarming that the word "rotation" does not occur in this manuscript. Pay attention to rotational questions.

There are certain weaknesses in the plan of this work, such as assuming that the rotational status of the original PMF model of X was correct or best possible (see below). These weaknesses cannot be corrected in an enhanced ms but they should be briefly discussed. This is important because otherwise, colleagues following the example of this work will feel the need to replicate everything that was done here, being unaware that some details may not have been optimal.

Enhance figure captions so that readers do not need to guess what is shown. Have the enhanced ms proofread by colleagues. Check also the references. This ms illustrates, once again, how difficult it is to find ones own mistakes and typos.

Notation used in this review.

The notation "[X Z]" indicates here attached or joined matrices, i.e. placing X and Z side by side so that they form one larger matrix.

The notations G(X) and F(X) will indicate factor matrices (G and F) obtained from an individual PMF model of X only, and similarly G(Z) and F(Z) for Z only.

The left and right parts of F, when modeling [X Z], are denoted by F[Xz] and F[xZ].

[Figure]

Q(X) and Q(Z) indicate Q values from separate analyzes of X and Z.

Similarly, Q[Xz] and Q[xZ] denote Q sums computed over elements of X and over elements of Z in the joint analysis of [X Z]. Hence, Q[X Z] = Q[Xz] + Q[xZ].

Total weight of X means the sum of squares of X_ij/s_ij over X, where s_ij is the uncertainty assumed for X_ij. If both X and Z are equally important, and if X and Z are of different sizes, all s_ij reported for the larger matrix should be increased so that total weights of X and Z become approximately equal. This implies a deviation from the general principle of determining weights from std-dev of values.

Background

Before examining this manuscript in detail, it is necessary to discuss the model that it tries to solve and the problems that make this task difficult. It is known that PMF of combined matrices often leads to disappointing results, such that some factors only (or mainly) fit X while other factors only/mainly fit Z. Such result is worthless in cases where X and Z are caused by the same emission sources whose emission profiles should be determined for X and Z.

It is important to realize what advantages may be expected from the joint analysis of X and Z. Three Cases are possible: PMF models computed separately for X and for Z may be valid and rotationally unique for (A) both X and Z, (B) one of them (for X, say), or (C) neither one of them.

Case A: If individually computed factors G(X) and G(Z) are practically identical, then a straight-forward joint model is successful for this case. Then G_[X Z] = G(X) = G(Z). If G(X) and G(Z) are significantly different, however, then the joint model will fail, producing too large residual values and hence too large Q. Such result might be caused e.g. by "common-mode errors" (see below) in X and/or in Z.

Case B: Now a joint model should be specified so that total weight (see Notations, above) of better-analyzed matrix X is significantly higher than total weight of Z. Then X

[Figure]

will "drive the model", and G_[X Z] will be approximately equal to G(X). If a reasonable Q[xZ] is obtained, then it indicates that X and Z are compatible, i.e. a joint PMF model is meaningful. Larger Z residuals and larger Q[xZ] would be obtained e.g. if X and Z do not have common sources or if there are common-mode errors. Then the joint PMF model is not meaningful for the chosen number of factors.

Case C: individual PMF models of both X and Z contain rotational ambiguity and/or other problems such as unidentifiable factors or missing factors. In this case, the approach of Case B cannot be used because the obtained ambiguous rotation, based mostly on X, may not be the best rotation for fitting Z. Ideally, equal total weights should be applied on X and Z, hoping that the best rotation for fitting both will be obtained when rotational information from Z is combined with information from X. Experience shows that quite often, such modeling fails. Few, if any, studies have been made about the reasons of such failures. It must be stressed that these failures must not be ascribed to "different units used in X and Z" (see below). As a first remedy, one might inspect the residuals in order to see if common mode errors are visible. Such errors might be corrected by hand, or by using an enhanced PMF model that automatically corrects for common mode errors. One might also inspect individual variables in order to see if only few variables are causing incompatibility of X and Z. Such variables might be downweighted in order to obtain a better overall model. Of course, one must also consider the possibility that in addition to their joint sources, X and Z may also have one or several unique sources. An enhanced PMF model may be developed for analysing such joint matrices containing common and non-common sources.

Summary of Case C: too little is known about reasons why this case fails. Well-documented case studies are needed. Singular value decompositions of G matrices computed for X, Z, and [X Z] may be useful for demonstrating the root of the problem. Reliable remedies may only be suggested when more is known about the reasons for failures in joint PMF modeling.

Common mode errors

Certain problems in measurements will cause so-called "common mode" errors. E.g. an error in air volume control in an aerosol sampler, when measuring sample i, causes that all aerosol concentrations on row i of X will change by the same fractional amount. Such common mode deviation does not contribute to residuals in customary PMF analysis of such aerosol data. Instead, common mode disturbance of sample i will change all elements of row i of matrix G. In a combined matrix, the other part Z is often measured using another instrument. Then Z may have its own common mode errors, different than those of X. In a joint analysis of X and Z, two independent sets of common mode errors will cause increased residuals when factors are common to X and Z. It appears highly probable that such common mode errors are an important reason for those PMF results where individual factors tend to fit either X or Z but not both.

Joint matrices containing different units

This ms claims that quantitative PMF modeling of a joint matrix [X Z] is not possible if variables in X and Z are measured in different units, such as mass concentration (expressed in mass/airvolume) and particle number concentration (expressed in particles/airvolume). These claims are based on a widespread misunderstanding, as explained in this section.

Customary aerosol PMF models are often scaled so that the sum of all elements in each row of matrix F equals unity. Then factor element $F\_pj$ indicates the fraction of species j in profile of source p. With joint matrices containing different units, summation over a row of F is not meaningful. The following workflow should be used instead in order to preserve the quantitative nature of the model:

In PMF (or after PMF), scale factors so that the average of each column of G is scaled ("normalized") to unity. Then elements of F have the following quantitative meaning: $F\_pj$ indicates the average contribution of source p to observations in column j, both for species j in matrix X and for species j in Z. The average total amount of all aerosol species in source p is obtained by summing values $F\_pj$ over all species j in F[Xz], i.e.

in the part of F corresponding to aerosol matrix X. In this way, the customary interpretation of F_pj as fractions of total may be obtained "off-line" after PMF computations by dividing the F_pj values by their sums taken over F[Xz].

The ms also suggests that presence of other variables (Z) in PMF model somehow makes the model non-quantitative or unreliable:

ms lines 79-80: there can be no confidence as to whether the sources are apportioned by units of number concentration (1/cm3) or any of the other units used in the auxiliary data.

Units may be entirely ignored in PMF modeling if all variables are represented in same units. If different units are present in different columns of matrix X, then the following practice is followed: elements of factor matrix G are pure numbers. Elements in column j of factor matrix F carry the same dimension and unit as column j of data matrix X. In the present case, all elements of left part F[Xz] of factor matrix F will be in mass/airvolume (same as X) while all elements of the right part F[xZ] are in units of number concentration (1/cm3) (same as Z). There is no confusion regarding dimensions or units.

Disturbance of quantitative modeling of X by "other variables" in Z may only be present if Z variables make the fit of X extremely poor, so that Q[Xz] increases to unacceptable levels in comparison to the original Q(X). This can be seen from Eq. (1) which defines PMF model: all values in column j of X are fitted using F factor elements from column j of F only. The "other columns" in F, corresponding to "other variables" in Z, do not enter in the fit of any X variables.

If Q[Xz] remains normal, model of X remains quantitative even when Z is introduced in modeling. However, if introduction of Z requires that number of factors must be increased, then the two models are different. Then rotational uniqueness and interpretatability of the joint model of [X Z] may well be better or worse in comparison to the original model of X only.

On the other hand, G(X) and G[X Z] may appear significantly different even when all Q values are normal. In this sense, including Z may interfere with the fit of X although the new fit of X remains as quantitative (or better) than the original fit of X. Such effect depends on rotational ambiguity of the original PMF fit of X: when Z is introduced, it may "rotate" a rotationally ambiguous model of X so that Z obtains a better fit while Q[Xz] does not increase from Q(X) or increases a little. Such rotation may only occurr if the original model of X is rotationally ambiguous, "non-quantitative". If such ambiguity is not understood by the scientist, it might appear that introduction of other variables "harms" the original model. In contrast, however, modifying the original model of X by a rotation is what is desired when using the joint model: both X and Z should be fitted as well as possible. This effect does not harm the quantitative nature of the model, as long as Q value of X does not grow too much.

Summary of this section: if Q computed over X elements increases significantly when modeling [X Z] instead of X, this indicates that X and Z are not compatible (when assuming this number of factors). Then analysis of [X Z] should be rejected. In all other cases, the joint model of X is equally good or better than the original model of X. If original model is rotationally ambiguous, then factors usually change: G[X Z] is different from G(X) and similarly F[Xz] is different from F(X). These new factors fit X as well as the original factors, thus they are as quantitative as the original factors. The rotation of these new factors takes into account information from matrix Z. In some cases, the new factors are rotationally unique, without any ambiguity. More often, the ambiguity of new factors is less than the original ambiguity.

Discussion of the manuscript

This manuscript suffers badly from almost complete avoidance of equations and mathematical symbols and mathematical notation in general. Also, there are serious problems in the few equations that are present. A more compact and easier to read presentation is obtained if mathematical notation is used as the primary means of communication. It is possible that part of my criticism in this review is simply based on

misunderstanding unclear and/or ambiguous verbal explanations of mathematical concepts.

The ideal of scientific work is repeatability. This ms does not provide facts that might enable repeatability, even in principle. E.g., I could not find dimensions of data matrices or obtained Q values. How were NSD data preprocessed before PMF computations? Using averages or medians? How were outliers handled? How many factors were used in each case? And so on.

The basic assumption of factor analytic modeling is that for each source, chemical profile and size distribution stay constant throughout the measurement campaign. On the other hand, it is well known that whenever nucleation happens, aerosol size distributions do vary. Also, largest particles tend to settle down more during longer transit times. In this work, constancy of size distributions was silently assumed. It might be good to discuss this fundamental question in future versions of this work.

Two-stage PMF model vs. customary PMF model

In the present ms, the goal was to determine the size distributions corresponding to the previously determined aerosol composition sources. It was assumed (on what grounds?) that the rotation of the original PMF result was correct, so that the originally obtained G matrix was deemed suitable for the PMF model of NSD matrix Z. In other words, it was desired that X "drive" the modeling of [X Z]. Essentially, this method corresponded to Case B, discussed above. Apparently, the authors were unaware of the one-stage method suggested for Case B. In hindsight, the best approach might have been to follow both Case B and Case C, especially if there was no positive information confirming that the original PMF model of X was rotationally unique and correct. An enhanced version of the ms should briefly discuss the one-stage possibilities of doing this work according to Case B and/or Case C. The one-stage method, with suitably weighted X and Z, would be easier to explain and much easier to understand. However, it is not reasonable to expect that the work be redone using the one-stage approach.

I understand step 2 so that the computed G factors from step 2 were forced to be practically identical to G factors from step 1. Is this right? If this is right, then step 2 appears to be equivalent to non-negative weighted regression (non-negative weighted linear least squares fit) of matrix Z by columns of matrix G. This should be mentioned. There are easy-to-use computer programs for computing such LS fits. Although PMF may also be used for this fit, using simpler tools would make the process more transparent, so avoiding unnecessary complications. Equations for defining the hidden factor should be given. The verbal definition is hard to understand and I did not manage to understand it.

The hidden factor, aka Nucleation factor

It is a good idea to assume that due to its higher time resolution, the NSD matrix Z may contain factors that are not visible in matrix X of chemical profiles. Unfortunately, the method for defining the hidden factor(s) in Z is questionable. First of all, why did you assume that there is only one hidden factor?

It seems that in stage 2, 6 factors were used. This is not defined (why not) but this is how I understand the ms. Why did you not use in 2nd stage PMF a 7th (and maybe an 8th) factor that may only fit the NSD part of the data matrix? This simple arrangement would determine hidden factor(s) avoiding the bias that non-negativity constraints may introduce in your method (see below). This alternative must be mentioned in a future version of the paper.

The second Equation (3) is incorrectly formulated. Symbol j is used as a summation index on the right side. Then it cannot appear on the left side. There is a symbol "x". It is not defined, what does it mean? The text says: "The Cran R package Non-Linear Minimization (nlm) (R Core Team, 2018) was used to minimise equation 3." You must not say "minimize equation". You must specify the expression that is minimized, and also specify the free variable(s) that are varied in order to minimize. I cannot understand the expression to minimize nor the free variables. For this reason, I cannot

comment more on determining the hidden factor. Maybe it is properly determined, maybe not. This part of the work is certainly not reproducible by others.

Bias: It seems that the second Equation (3) is not applied to all data because of non-negativity constraints (however, there seems to be an error in the constraints, it is impossible to guess what was really intended). When some data are excluded, this creates a bias. It is impossible to know from the outside if this bias was negligible or if it distorted the results. The bias question must be documented.

Miscellaneous

Lines 415-417 in Conclusion: "This generates confidence that the NSD and PM10 factors ascribed to one source are in fact attributable to that same source." This is a very important statement, good!

There are two equations numbered (3). This caused a LOT of trouble when trying to understand the discussion of the "hidden profile" a.k.a. "nucleation profile". The first Equation 3 does not appear correctly on my computer. Possibly, it uses a symbol font that is not present on my computer so that one symbol is not visible. There is also another problem in this equation: symbol "a" is used as summation index, and symbol "a" appears also on left side. A summation index cannot be present on left side. Please check your equations before submitting new versions of the ms. Make sure that the .pdf file contains all non-standard fonts that are used e.g. in equations.

The presentation should be helpful for the reader. The symbols used in text and in equations should be defined. Example: in first Eq. (3), there is symbol j. What does it mean? Is it the index of size bin? Why not help the reader and say so? In second Eq. (3), there is again a symbol j. What is it now? Please update the ms so that symbols are used in a systematic way, in order to help the reader. The following method is recommended in order to avoid confusion with symbols:

For your own use, create a table where each symbol, however trivial, is entered. When

needing more symbols, check first with the table if the symbol is already reserved for another use. When you are ready, include short definitions from the table into the ms, either in a table of notation or to the location of first use of each symbol. Use customary matrix element notation whenever possible. In this way, you could avoid using scalar "a" first as an index and then vector a_j as a vector of unknowns.

Description of the linear regression model (section 2.4) is strange. I have never seen that the coefficients are called "gradients". Also, correlations should not be mentioned when discussing linear least squares. It would be best to simply show the equation. I recommend that explanation of regression be omitted, except that the equation, using matrix element notation, should be shown.

Figure 6 is unclear. What is illustrated by the bivariate plots? Figure caption only tells that they are bivariate plots, plotted using the Openair program. Instead of naming the plotting program, it would be more important to define what is plotted vs. what, and what are the dimensions in individual diagrams. After working with the ms for a long time, I tend to guess that the "bivariate plots" might represent NSD concentrations in polar plots of wind direction and wind speed. Why did you not say this? Saving one sentence from the ms may cost hours for your new readers.

---

## Referee Comment (RC3) · Anonymous Referee #3 · 7 Nov 2018

The manuscript "RECEPTOR MODELLING OF BOTH PARTICLE COMPOSITION AND SIZE DISTRIBUTION FROM A BACKGROUND SITE IN LONDON, UK – THE TWO STEP APPROACH" presents a new approach (based on 2 steps) for source apportionment studies using positive matrix factorization (PMF). This method aims to properly handle dataset(s) composed of variables with multiple units (mass and number concentration, in this case). The authors claim that this new method overcomes the problem of inputting variables with heterogeneous measurement units. The authors also claim that the method is able to better detect hidden factors/sources.

The manuscript has several elements of originality (to my knowledge, no similar methods have been already published) and directly hits a very controversial and up-to-date topic in the atmospheric sciences. Nowadays, source apportionment by PMF is amply used in both routine monitoring and research studies. Although most of them use "one-kind" variables (mostly PM chemical speciation data), an increasingly high number of studies (just a few have been cited in the manuscript, but the list should be improved) use variables with multiple units. Since the large number of available air quality measurement techniques, the merging of dataset(s) with different units is a suitable (and proven) way to better resolve the PMF source profiles and to detect unresolved sources. Essentially, additional variables may help in better detecting the edges. Under this view, a recent paper (Emami and Hopke, Chemometr. Intell. Lab. 162 (2017) 198–202, which findings are unfortunately not considered in this manuscript), showed the effect of adding variables with different units to decrease the rotational ambiguity of PMF solutions.

Thus, the topic is suitable for the journal ACP. However, the manuscript needs revisions before to be accepted for publication.

Major points.

Essentially, the rationale behind the whole manuscript is based on the statement reported in lines 59-62: "However, while combining, for example, particle chemical composition and size distribution data in a single PMF analysis may assist source resolution, it does not allow quantitative attribution of either particle mass or particle number to the source factors.". Later, the authors also presented a case study where they mixed variables with different units without giving quantitative results. Even if one can agree with this statement, the authors have not exhaustively explained it. Since this is a methodological manuscript, I strongly encourage the authors to better support these statements.

Another major weakness of this manuscript is the lack of sufficient details on the PMF

analyses. This point can be easily solved by the authors, who have an extended experience with PMF analysis. This manuscript presents a new approach, so particular care should be given to details so that anyone can easily reproduce what the authors did (and test with their own data). However, details of the PMF are generally missing or they are reported in the companion paper (Beddows et al., 2015). For example, the authors should describe the method(s) used to compute the uncertainties in the 1st step (including PN0.6-10, see next point). Also, the authors should report how the raw data have been handled (if any correction was done) and the number of variables and cases inputted into the models. For example, they should report the outliers detection and how they managed the missing values (SMPS sampled every 15 min, what is the minimum number of 15 min records to have a valid 1-hour NSD value?). In addition, since the Q values are used (lines 199-203 and 215-216), they should be reported as well. Furthermore, it is unknown if the authors dealt with the rotational ambiguity of the models. Basic information on the PMF set-up is important to report. This information will allow the reader to completely understand what the authors did and (possibly) to reply the methods. It would be useful to have a quick overview of such details in the main text with the deepest description in the supplementary information.

Another unclear point is related to the "proxy-data" used to assess the PN0.6-10 variable. This is an artificial variable: it was not directly measured, but it was computed on the basis of two (three?) main assumptions: (i) particles are assumed to be spherical, and (ii) particles have fixed density. But it is not completely clear if the density is assumed constant over the time or over the whole (16-604 nm) size spectra (or both, as it should be). The authors used a density of 2 g/cm3 over all the study period, but they report a 1.8-2.5 g/cm3 range for an urban background aerosol. Consequently, the PN0.6-10 variable will be affected by a large uncertainty that cannot be well assessed. I suggest to add more details and provide an estimate of the uncertainty of this new variable.

This latter point raises another question. Why the authors did not plan to also use an

[Figure]

APS to complete the size range to 10 $\mu$m? One can argue that the sampling campaign was not planned to have an APS included (or the merging of SMPS and APS was unreliable). However, my opinion is that this point should be at least mentioned in the text, so colleagues who want to pursue the same approach are advised on the possible use of wide range particle size spectra.

The authors used R to "optimize" X to have Q/Qtheory $\sim$ 1. More details should be reported. In particular, what does "$\sim$ 1" mean? It can be every number, but having it from 0.5 to 1.5 or from 0.99 to 1.01 makes a big difference. Please explain.

Minor comments.

Line 103. Missing bracket ")"

Subsection 2.1: Please add more details on the SMPS set-up. For example, sheath and sample flows, the status of the CPC and electrostatic classifier (serviced, calibrated?), the type of neutralizer (X-ray, 85Kr?), software/algorithm used for the data inversion (or version of the AIM software), use of multiple charge and/or diffusion loss corrections, etc. These details need to be added as supplementary information.

Line 138: $\frac{1}{4}$ hour -> 15 min

There are two equations numbered as (3), see pages 11 and 12. This should be fixed, as most of the discussion on the method refers to these equations.

Figure 3 can be easily moved to the supplementary material file.

Figure 4. NET & coarse should be NET & crustal.

Figure 6. Once printed, the labels and axes of the single plots will be likely unreadable. Please increase the font size and (if possible) please uniform the font style and size among the figures. Also, it is advisable to use a color scale that is also easily readable when the paper is printed with a black and white printer.

Figure 6 shows polarplots and polarannuli. These "openair" analyses are commonly

reported in air quality studies and are very helpful to better interpret the data. However, a quick overview of the information provided by these two plots should be briefly reported into the materials and methods section.

---

## Referee Comment (RC4) · Anonymous Referee #4 · 8 Nov 2018

The paper by Beddows et al. described a two-step source apportionment methodology on a combined database of both PM mass and number size distribution measurements carried out in London. A previous source apportionment study using the same database had been reported by Beddows et al. (2015). Thus, the novelty of this study could be represented by the methodology development. The topic is interesting, and the methodology would be useful in deal with mixing data types as input in PMF, which provide a better defined source factor and better fit diagnostics compared to when non-combined data were used. However, I found that some aspects are not clear and

improvements should be made before the work be published in ACP.

Major comments: 1. The motivation of this study is to clarify the source contribution when a combined database was used in PMF. As the authors state, the combined PM chemical composition and size distribution data in a single PMF analysis could not allow quantitative attribution of either particle mass or particle number to the source factors. However, one could calculate the source contributions either by PM mass or by NSD base on the output results of PMF. The following reference is an example described the source contribution using combined database in PMF. Please clarify this item. Sowlat et al., 2016. Source apportionment of ambient particle number concentrations in central Los Angeles using positive matrix factorization (PMF). Atmos. Chem. Phys., 16, 4849-4866,

2. The two-step PMF-PMF method is new but the results maybe questionable. The G1 time series from the PMF analysis of PM10 chemical composition (Step One) could be considered as a constraint in Step Two, which means that six factors identified by PM mass was also applied to NSD. I think this is why the results from two step PMF-PMF method was different from results using combined dataset of PM and NSD in PMF reported by Beddows et al. (2015). Thus, what about the results if using the G1 time series from the PMF analysis of NSD as step one? Please clarify this item.

Specific comments 1. Line 157-160. The particle number greater than 600nm is calculated from the difference between PM10 and PM0.6 estimated from SMPS. Except PM0.6-10, particle density, particle shape (spherical) and size distribution should be know when calculated the PN0.6-10. Please provide more description about the calculation process. 2. Line 355-356. Why the secondary factor be expected to be strongest at night? 3. Line 362-363. These is not Fig.7 in the text.
* * *

---

## Author Comment (AC1) · 16 Jan 2019

RESPONSE TO REVIEWERS

REVIEWER #1 General comment: The paper regards the description of a two-step approach for performing source apportionment using the PMF receptor model and an input composed by variables having different measurement units. The approach has elements of originality and potentially several applications. The topic is interesting considering that source apportionment is a major topic in nowadays research and the

possibility to use an approach that use input variables having heterogeneous measurement units is certainly appealing. I also believe that the topic is suitable for the Journal and the paper generally well written and understandable. However, I found that some aspects are not completely clear (see my specific comments) and the paper need a revision before publication. RESPONSE: We thank the reviewer for these positive overall remarks.

Speciﬡc comments: Lines 59-62. This sentence is not completely true and I would suggest to modify it. This happens only if number size distributions are mixed with chemical composition (in mass), however, there are examples in which size-segregated chemical composition is used in PMF analysis to obtain quantitative evaluation of size distribution of sources (see for example Contini et al., 2014 Science of the Total Environment 472, 248–261 and references therein). RESPONSE: Text has been added to this effect, saying that by careful experimental design the issue of datasets with heterogeneous units can be avoided, for example using a Cascade Impactor to measure size-fractionated chemical PM mass composition rather than two measurements: one for particle number size and the other for total PM mass.

Lines 140-141. Better to write 16-604 nm (like in line 149) because two decimal digits for size is an illusory precision. RESPONSE: Correction made.

Lines 159-161. The conversion of mass difference in PN0.6-10 is likely quite uncertain. Some details should be given because I believe that some assumptions have been done regarding size distribution in the range 0.6-10 micron and the result of the conversion would be strongly influenced by these assumptions. A comment on this aspect is needed. RESPONSE: This section has been re-written to include more details and a statement added that a large uncertainty is applied to this measurement so as not to influence the final results.

Lines 312-314. This could happen because nanoparticles have a limited mass to influence significantly PM10 mass composition, however, it could be different if NSD

are mixed with PM1 chemical composition for example. A comment on this aspect would be useful. RESPONSE: Within the context of response to comments on lines 59-62, we have commented on this to say that with a different measurement, PM1, the NSD data would give a better overlap. However, having said this, PN measurements have a sensitivity bias towards the smaller nucleation particles whereas PM measurements have a bias towards the more coarse particles.

There is a particular reason for using PMF2 and not the more advanced PMF5 that is becoming the standard version of source apportionment with PMF? RESPONSE: PMF2 is not version 2 of the US EPA PMF. PMF2 is the ordinary 2-way factor analysis as opposed to the 3-way factor analysis PMF3 or ME-2 for solving arbitrary (quasi) multilinear models. This has been clarified in the Experimental part of the text.

Lines 215-216. How much is it the X value chosen? This should likely be reported for completeness. RESPONSE: We have added this information, which says that once fitted the NSD data have a relative uncertainty of 4-5%.

Line 259. I believe that the number of factors is six rather than seven. RESPONSE: Yes, the correction has been made.

Lines 358-360. Looking at figures 4 and S3, it seems that the marine source is dominated by nanoparticles. Considering that this is a source generally made of coarse particles, and also authors mention this aspect, this result appears unusual and some discussion and explanations are needed. RESPONSE: This apparent contradiction has already been addressed in Beddows et al. (2015) in the five-factor solution from the combined composition–NSD data set. In this, a factor which can be clearly assigned on the basis of its chemical association is that described as aged marine. This explains a large proportion of the variation in Na, Mg and Cl but shows a NSD with many features similar to that of the traffic factor, with which it has rather little in common chemically. Since the aged marine mass mode is expected to be in the super-micrometre region and hence well beyond that measured in the NSD data set, it seems likely that the size

distribution associated is simply a reflection of other sources influencing air masses rich in marine particles. The main point to take away is that we get the same solution using the 2-step approach.

Minor comments: Lines 148. "spherical" RESPONSE: Yes, the correction has been made.

Line 351. "there is..." The source "NET and crustal" is reported in the text but repeated in the figures as "NET and coarse". I would suggest to use "NET and crustal" in all the paper that is more understandable and appropriate. RESPONSE: Yes, the correction has been made to be consistent with our original work in Beddows et al. (2015).

Title Section 3.4. Why hidden? RESPONSE: The work hidden has been replaced by unresolved. It was not resolved until a 7 factor solution was chosen using an FKEY matrix (as specified in Figure 3 with 6 x 6 zero diagonal FKEY matrix augmented with an 7th column and 7th row of zero entries).

Moreover, this section is dedicated to several factors...what is the hidden one the nucleation? An explanation or a change of the title is needed. RESPONSE: Extra explanation is given.

What is the meaning of the "*" reported in figures 4 and 5? RESPONSE: These have been removed.

REVIEWER #3 The manuscript has several elements of originality (to my knowledge, no similar methods have been already published) and directly hits a very controversial and up-to-date topic in the atmospheric sciences. Nowadays, source apportionment by PMF is amply used in both routine monitoring and research studies. Although most of them use "one-kind" variables (mostly PM chemical speciation data), an increasingly high number of studies (just a few have been cited in the manuscript, but the list should be improved) use variables with multiple units. Since the large number of available air quality measurement techniques, the merging of dataset(s) with different units is a

suitable (and proven) way to better resolve the PMF source profiles and to detect unresolved sources. Essentially, additional variables may help in better detecting the edges. RESPONSE: We thank the reviewer for this positive perspective on the work.

Under this view, a recent paper (Emami and Hopke, Chemometr. Intell. Lab. 162 (2017) 198–202, which findings are unfortunately not considered in this manuscript), showed the effect of adding variables with different units to decrease the rotational ambiguity of PMF solutions. RESPONSE: This paper has now been cited.

Thus, the topic is suitable for the journal ACP. However, the manuscript needs revisions before to be accepted for publication. Major points. Essentially, the rationale behind the whole manuscript is based on the statement reported in lines 59-62: "However, while combining, for example, particle chemical composition and size distribution data in a single PMF analysis may assist source resolution, it does not allow quantitative attribution of either particle mass or particle number to the source factors.". Later, the authors also presented a case study where they mixed variables with different units without giving quantitative results. Even if one can agree with this statement, the authors have not exhaustively explained it. Since this is a methodological manuscript, I strongly encourage the authors to better support these statements. RESPONSE: The comment regarding the unapportioned factor analysis of data with heterogeneous units from the supporting output of Beddows et al. (2015), is referred to in Section 3.2 entitled 2-Step PMF-LR Analysis. We have expanded this section of text to report more clearly, what was carried out in the supporting study.

Another major weakness of this manuscript is the lack of sufficient details on the PMF analyses. This point can be easily solved by the authors, who have an extended experience with PMF analysis. This manuscript presents a new approach, so particular care should be given to details so that anyone can easily reproduce what the authors did (and test with their own data). However, details of the PMF are generally missing or they are reported in the companion paper (Beddows et al., 2015). • For example, the authors should describe the method(s) used to compute the uncertainties in the

1st step (including PN0.6-10, see next point). RESPONSE: This section has been rewritten to give more detail.

â˘Ać Also, the authors should report how the raw data have been handled (if any correction was done) and the number of variables and cases inputted into the models. For example, they should report the outliers detection and how they managed the missing values (SMPS sampled every 15 min, what is the minimum number of 15 min records to have a valid 1-hour NSD value?). RESPONSE: The details of the SMPS setup are now in Table S1 and a note is added to say that the raw data was quality assured by the National Physical Laboratory (NPL), and to see Beccaceci et al. (2013a,b) for an extensive report on how the data was collected. Furthermore, we addressed these points by carrying forward the descriptions in Beddows et al. (2015) of how the PM10 data was collected and prepared for this study.

In addition, since the Q values are used (lines 199-203 and 215-216), they should be reported as well. Furthermore, it is unknown if the authors dealt with the rotational ambiguity of the models. RESPONSE: This is addressed in the response to P. Paatero's comments.

The authors used R to "optimize" X to have Q/Qtheory âĹij 1. More details should be reported. In particular, what does "âĹij 1" mean? It can be every number, but having it from 0.5 to 1.5 or from 0.99 to 1.01 makes a big difference. Please explain. RESPONSE: We have set a criterion of within $1 \pm 0.02$.

Basic information on the PMF set-up is important to report. This information will allow the reader to completely understand what the authors did and (possibly) to reply the methods. It would be useful to have a quick overview of such details in the main text with the deepest description in the supplementary information. RESPONSE: This point has been addressed within the rewritten mathematical description of the PMF analysis.

Another unclear point is related to the "proxy-data" used to assess the PN0.6-10 variable. This is an artiïïñĄcial variable: it was not directly measured, but it was computed

on the basis of two (three?) main assumptions: (i) particles are assumed to be spheri-
cal, and (ii) particles have fixed density. But it is not completely clear if the density is
assumed constant over the time or over the whole (16-604 nm) size spectra (or both,
as it should be). The authors used a density of 2 g/cm3 over all the study period, but
they report a 1.8-2.5 g/cm3 range for an urban background aerosol. Consequently, the
PN0.6-10 variable will be affected by a large uncertainty that cannot be well assessed.
I suggest to add more details and provide an estimate of the uncertainty of this new
variable. RESPONSE: Clarification of this has been made by adding a fuller and more
mathematical description to explain how the proxy variable is calculated and how the
density value is used.

This latter point raises another question. Why the authors did not plan to also use an
APS to complete the size range to 10 $\mu$m? One can argue that the sampling campaign
was not planned to have an APS included (or the merging of SMPS and APS was
unreliable). However, my opinion is that this point should be at least mentioned in the
text, so colleagues who want to pursue the same approach are advised on the possible
use of wide range particle size spectra. RESPONSE: We have added this point to a
list of alternative approaches to using the proxy-data at the end of Section 2.2.

Minor comments: Line 103. Missing bracket ")" RESPONSE: Corrected.

Subsection 2.1: Please add more details on the SMPS set-up. For example, sheath
and sample flows, the status of the CPC and electrostatic classifier (serviced, cal-
ibrated?), the type of neutralizer (X-ray, 85Kr?), software/algorithm used for the data
inversion (or version of the AIM software), use of multiple charge and/or diffusion loss
corrections, etc. These details need to be added as supplementary information. RE-
SPONSE: This has been added in Table S1, although it does seem like too much
information for what the referee correctly identifies as a PMF methodology paper; it is
not a data collection paper.

Line 138: 1 4 hour -> 15 min RESPONSE: Change made although this is considered

to be a personal preference.

There are two equations numbered as (3), see pages 11 and 12. This should be fixed, as most of the discussion on the method refers to these equations. RE-SPONSE: Correction made.

Figure 3 can be easily moved to the supplementary material file. RESPONSE: Figure moved.

Figure 4. NET & coarse should be NET & crustal. RESPONSE: Correction made.

Figure 6. Once printed, the labels and axes of the single plots will be likely unreadable. Please increase the font size and (if possible) please uniform the font style and size among the figures. Also, it is advisable to use a color scale that is also easily read-able when the paper is printed with a black and white printer. RESPONSE: We have increased the font size at the expense of the size of the plots which has improved the readability of the text in these plots. However, we have not found a palette which looks good in colour and preserves the information in black and white. All we can suggest is that a grey scale is used for the option of black and white printing.

Figure 6 shows polarplots and polarannuli. These "openair" analyses are commonly reported in air quality studies and are very helpful to better interpret the data. However, a quick overview of the information provided by these two plots should be briefly reported into the materials and methods section. RESPONSE: General descriptions of polarPlot and polarAnnulus have been added to the Methods Section.

REVIEWER #4 The paper by Beddows et al. described a two-step source appor-tionment methodology on a combined database of both PM mass and number size distribution measurements carried out in London. A previous source apportionment study using the same database had been reported by Beddows et al. (2015). Thus, the novelty of this study could be represented by the methodology development. The topic is interesting, and the methodology would be useful in deal with mixing data types

as input in PMF, which provide a better defined source factor and better fit diagnostics compared to when non-combined data were used. However, I found that some aspects are not clear and improvements should be made before the work be published in ACP. RESPONSE: We thank the referee, and we welcome the opportunity to provide greater clarity.

Major comments: 1. The motivation of this study is to clarify the source contribution when a combined database was used in PMF. As the authors state, the combined PM chemical composition and size distribution data in a single PMF analysis could not allow quantitative attribution of either particle mass or particle number to the source factors. However, one could calculate the source contributions either by PM mass or by NSD base on the output results of PMF. The following reference is an example described the source contribution using combined database in PMF. Please clarify this item. Sowlat et al., 2016. Source apportionment of ambient particle number concentrations in central Los Angeles using positive matrix factorization (PMF).Atmos. Chem. Phys., 16, 4849-4866, RESPONSE: We do calculate the source contributions either by PM mass or by NSD based on the process of using output from the PMF results in Beddows et al. (2015). Those results are carried through into this work, so we are already carrying out a 1-step analysis resulting in an apportionment. To address this oversight of the referee, we have added a line to Figure 2 saying "[The PMF analyses of Beddows et al. (2015) are considered as Step 1]." We have also added a table of apportionment values from Beddows et al. (2015) into Figure 1 as an insert showing the apportionment of the factors, and the reference to Sowlat et al. (2016) which is very similar to Harrison et al. (2010), which reports PMF of merged SMPS-APS data and chemical and meteorological data.

2. The two-step PMF-PMF method is new but the results maybe questionable. The G1 time series from the PMF analysis of PM10 chemical composition (Step One) could be considered as a constraint in Step Two, which means that six factors identified by PM mass was also applied to NSD. I think this is why the results from two step

PMF-PMF method was different from results using combined dataset of PM and NSD in PMF reported by Beddows et al. (2015). RESPONSE: The reviewer is correct in this interpretation.

Thus, what about the results if using the G1 time series from the PMF analysis of NSD as step one? Please clarify this item. RESPONSE: The aim here is to assign a NSD description to the PM10 mass sources, so we are not sure why we would consider a 1-step PMF analysis of the combined G1 + NSD data set without applying an 'FKEY constraint'. When removing the FKEY constraint, there is no clear separation of the G1 scores and we can no longer match the NSD of the resulting factors to the original source. Instead we have to introduce new descriptions based on the 6 factor names: Diffuse Urban; Marine; Secondary; NET / Coarse; Fuel Oil and Traffic. Furthermore, a conclusion from Beddows et al. (2015) was that a better result was obtained when analysing the datasets separately. This work continues with this recommendation by heavily biasing the analysis to the data analysed in Step 1.

Specific comments: 1. Line 157-160. The particle number greater than 600nm is calculated from the difference between PM10 and PM0.6 estimated from SMPS. Except PM0.6-10, particle density, particle shape (spherical) and size distribution should be know when calculated the PN0.6-10. Please provide more description about the calculation process. RESPONSE: This point has been address in line with the comment of Referee #1.

2. Line355-356. Why the secondary factor be expected to be strongest at night? 3. Line 362-363. These is not Fig.7 in the text. RESPONSE: Typo: Figure 7 is Figure 6. This has been corrected. Furthermore, the secondary factor is expected to be stronger at night when compared to the secondary NSD factor derived in Beddows et al. (2015). In Beddows et al. (2015), both the secondary component derived from the PM10 and NSD analysis are strongest at night, and in particular, the PM10 secondary factor has a strong nitrate component which does grow to a maximum during the night due to reduced volatility of ammonium nitrate. Clarification has been given.

REFEREE: P. PAATERO pentti.paatero86@gmail.com This manuscript deals with PMF analyses of "combined" data matrices such as [X Z] where X contains elemental composition profiles of aerosol samples and Z contains aerosol number size distributions measured simultaneously with composition profiles. This is an important problem that occurs often in modern aerosol research. There are specific problems in this task; these problems have not been studied in depth in literature so far. This manuscript studies one specific combined data matrix and reports a PMF model for this matrix. Thus the ms might deserve publication despite of certain serious problems. These problems are in part related to misunderstandings found in earlier papers that discuss this same topic. For this reason, the present review contains a lengthy general discussion of the task of modeling combined matrices. The specific questions regarding this ms are based on this general discussion. The ms might also be suitable for publication in the sister Journal AMT, tmospheric Measurement Techniques. My personal view is slightly in favour of AMT. However, both ACP and AMT seem possible, and this review considers publication in either Journal. The structure of this review is as follows: RESPONSE: We recognise the immense contribution made by Professor Paatero to this field, and thank him for the critical insights which he provides.

=============================================== Recommendations Notation used in this review Background Common mode errors Joint matrices containing different units Discussion of the manuscript Two-stage PMF model vs. customary PMF model The hidden factor, aka Nucleation factor Miscellaneous

Recommendations There are very many problems of different kinds in this manuscript. For this reason, I hesitantly recommend that this ms should NOT be published by ACP or AMT. However, if it is desired to publish this ms because of the importance of the problem, then a thorough rewriting of the text and mathematical details must be undertaken. I recommend that the following enhancements be performed: There has apparently been lack of communication between the person(s) who did the actual computations and those who wrote the paper. RESPONSE: This first author is responsible

for both the computation and the manuscript and we have endeavoured to follow these recommendations to avoid the appearance that an unnamed contributor has been involved. The paper has been extensively revised in response to the comments of all four reviewers. For this reason, the mathematical description is erratic, chaotic and impossible to understand or replicate. In order to create an accurate description, the person(s) who did the computations should be included in the group of authors. Without such help, it may not be possible to achieve a satisfactory mathematical description of what was done. The entire mathematical discussion about problems attributed to PMF analysis of joint matrices containing different units is erroneous, based on a widespread misunderstanding. This discussion must be rewritten according to suggestions given below. It might be good to include in the author group somebody familiar with the quantitative mathematical structure of the PMF model. RESPONSE: Thank you for highlighting this misunderstanding which we have addressed in the revised manuscript.

In particular, it seems that lines 79,80 are not based on quantitative understanding of the model. These lines, and other similar sentences, must be removed. Much of Conclusions must be rewritten so that the claims against using variables with different dimensions/units are replaced by opposite sentences stating e.g. that a joint analysis of matrices of variables with different dimensions/units is not harmed by these differences but unfortunately the opposite was believed to be true when the work was carried out. RESPONSE: This correction has been carried out.

The mathematical description of what was done must be totally rewritten so that systematic matrix-form notation is used. Equations must be corrected and written in correct notation, using correct terminology and correct numbering. Details of PMF modeling must be reported, such as dimensions of matrices, used parameters such as uncertainties of data values, robust/nonrobust, obtained Q values, numbers of observed outliers, unique or multiple minima, and so on. RESPONSE: This has been corrected following the guidance of all the other referees.

Rotational questions are an ever-present problem in factor analytic modeling, indepen-

dently of what programs are used. It is alarming that the word "rotation" does not occur in this manuscript. Pay attention to rotational questions. There are certain weaknesses in the plan of this work, such as assuming that the rotational status of the original PMF model of X was correct or best possible (see below). These weaknesses cannot be corrected in an enhanced ms but they should be briefly discussed. This is important because otherwise, colleagues following the example of this work will feel the need to replicate everything that was done here, being unaware that some details may not have been optimal. RESPONSE: Rotations are now briefly discussed.

Enhance figure captions so that readers do not need to guess what is shown. Have the enhanced ms proofread by colleagues. Check also the references. This ms illustrates, once again, how difficult it is to find ones own mistakes and typos. RESPONSE: Enhancements of figure captions have been carried out.

Notation used in this review The notation "[X Z]" indicates here attached or joined matrices, i.e. placing X and Z side by side so that they form one larger matrix. The notations $G(X)$ and $F(X)$ will indicate factor matrices (G and F) obtained from an individual PMF model of X only, and similarly $G(Z)$ and $F(Z)$ for Z only. The left and right parts of F, when modeling [X Z], are denoted by $F[Xz]$ and $F[xZ]$. $Q(X)$ and $Q(Z)$ indicate Q values from separate analyzes of X and Z. Similarly, $Q[Xz]$ and $Q[xZ]$ denote Q sums computed over elements of X and over elements of Z in the joint analysis of [X Z]. Hence, $Q[X Z] = Q[Xz] + Q[xZ]$. Total weight of X means the sum of squares of $X_{ij}/s_{ij}$ over X, where $s_{ij}$ is the uncertainty assumed for $X_{ij}$. If both X and Z are equally important, and if X and Z are of different sizes, all $s_{ij}$ reported for the larger matrix should be increased so that total weights of X and Z become approximately equal. This implies a deviation from the general principle of determining weights from std-dev of values. RESPONSE: An amended notation as suggested has now been implemented.

Background Before examining this manuscript in detail, it is necessary to discuss the model that it tries to solve and the problems that make this task difficult. It is known that PMF of combined matrices often leads to disappointing results, such that some

factors only (or mainly) fit X while other factors only/mainly fit Z. Such result is worthless in cases where X and Z are caused by the same emission sources whose emission profiles should be determined for X and Z. It is important to realize what advantages may be expected from the joint analysis of X and Z. Three Cases are possible: PMF models computed separately for X and for Z may be valid and rotationally unique for (A) both X and Z, (B) one of them (for X, say), or (C) neither one of them.

Case A: If individually computed factors G(X) and G(Z) are practically identical, then a straight-forward joint model is successful for this case. Then $G\_[X\ Z] = G(X) = G(Z)$. If G(X) and G(Z) are significantly different, however, then the joint model will fail, producing too large residual values and hence too large Q. Such result might be caused e.g. by "common-mode errors" (see below) in X and/or in Z.

Case B: Now a joint model should be specified so that total weight (see Notations, above) of better-analyzed matrix X is significantly higher than total weight of Z. Then X will "drive the model", and $G\_[X\ Z]$ will be approximately equal to G(X). If a reasonable Q[xZ] is obtained, then it indicates that X and Z are compatible, i.e. a joint PMF model is meaningful. Larger Z residuals and larger Q[xZ] would be obtained e.g. if X and Z do not have common sources or if there are common-mode errors. Then the joint PMF model is not meaningful for the chosen number of factors.

Case C: individual PMF models of both X and Z contain rotational ambiguity and/or other problems such as unidentifiable factors or missing factors. In this case, the approach of Case B cannot be used because the obtained ambiguous rotation, based mostly on X, may not be the best rotation for fitting Z. Ideally, equal total weights should be applied on X and Z, hoping that the best rotation for fitting both will be obtained when rotational information from Z is combined with information from X. Experience shows that quite often, such modeling fails. Few, if any, studies have been made about the reasons of such failures. It must be stressed that these failures must not be ascribed to "different units used in X and Z" (see below). As a first remedy, one might inspect the residuals in order to see if common mode errors are visible. Such errors might be

corrected by hand, or by using an enhanced PMF model that automatically corrects for common mode errors. One might also inspect individual variables in order to see if only few variables are causing incompatibility of X and Z. Such variables might be downweighted in order to obtain a better overall model. Of course, one must also consider the possibility that in addition to their joint sources, X and Z may also have one or several unique sources. An enhanced PMF model may be developed for analysing such joint matrices containing common and non-common sources.

Summary of Case C: too little is known about reasons why this case fails. Well documented case studies are needed. Singular value decompositions of G matrices computed for X, Z, and [X Z] may be useful for demonstrating the root of the problem. Reliable remedies may only be suggested when more is known about the reasons for failures in joint PMF modeling. RESPONSE: An account of cases B and C has been added to the paper.

Common mode errors Certain problems in measurements will cause so-called "common mode" errors. E.g. an error in air volume control in an aerosol sampler, when measuring sample i, causes that all aerosol concentrations on row i of X will change by the same fractional amount. Such common mode deviation does not contribute to residuals in customary PMF analysis of such aerosol data. Instead, common mode disturbance of sample i will change all elements of row i of matrix G. In a combined matrix, the other part Z is often measured using another instrument. Then Z may have its own common mode errors, different than those of X. In a joint analysis of X and Z, two independent sets of common mode errors will cause increased residuals when factors are common to X and Z. It appears highly probable that such common mode errors are an important reason for those PMF results where individual factors tend to fit either X or Z but not both. Joint matrices containing different units This ms claims that quantitative PMF modeling of a joint matrix [X Z] is not possible if variables in X and Z are measured in different units, such as mass concentration (expressed in mass/airvolume) and particle number concentration (expressed in particles/airvolume). These claims

are based on a widespread misunderstanding, as explained in this section. Customary aerosol PMF models are often scaled so that the sum of all elements in each row of matrix F equals unity. Then factor element F_pj indicates the fraction of species j in profile of source p. With joint matrices containing different units, summation over a row of F is not meaningful. The following workflow should be used instead in order to preserve the quantitative nature of the model: In PMF (or after PMF), scale factors so that the average of each column of G is scaled ("normalized") to unity. Then elements of F have the following quantitative meaning: F_pj indicates the average contribution of source p to observations in column j, both for species j in matrix X and for species j in Z. The average total amount of all aerosol species in source p is obtained by summing values F_pj over all species j in F[Xz], i.e. in the part of F corresponding to aerosol matrix X. In this way, the customary interpretation of F_pj as fractions of total may be obtained "off-line" after PMF computations by dividing the F_pj values by their sums taken over F[Xz]. The ms also suggests that presence of other variables (Z) in PMF model somehow makes the model non-quantitative or unreliable: ms lines 79-80: there can be no confidence as to whether the sources are apportioned by units of number concentration (1/cm3) or any of the other units used in the auxiliary data. Units may be entirely ignored in PMF modeling if all variables are represented in same units. If different units are present in different columns of matrix X, then the following practice is followed: elements of factor matrix G are pure numbers. Elements in column j of factor matrix F carry the same dimension and unit as column j of data matrix X. In the present case, all elements of left part F[Xz] of factor matrix F will be in mass/airvolume (same as X) while all elements of the right part F[xZ] are in units of number concentration (1/cm3) (same as Z). There is no confusion regarding dimensions or units.

Disturbance of quantitative modeling of X by "other variables" in Z may only be present if Z variables make the fit of X extremely poor, so that Q[Xz] increases to unacceptable levels in comparison to the original Q(X). This can be seen from Eq. (1) which defines PMF model: all values in column j of X are fitted using F factor elements from column j of F only. The "other columns" in F, corresponding to "other variables" in Z, do not enter

in the fit of any X variables. If Q[Xz] remains normal, model of X remains quantitative even when Z is introduced in modeling. However, if introduction of Z requires that number of factors must be increased, then the two models are different. Then rotational uniqueness and interpretatability of the joint model of [X Z] may well be better or worse in comparison to the original model of X only. On the other hand, G(X) and G[X Z] may appear significantly different even when all Q values are normal. In this sense, including Z may interfere with the fit of X although the new fit of X remains as quantitative (or better) than the original fit of X. Such effect depends on rotational ambiguity of the original PMF fit of X: when Z is introduced, it may "rotate" a rotationally ambiguous model of X so that Z obtains a better fit while Q[Xz] does not increase from Q(X) or increases a little. Such rotation may only occurr if the original model of X is rotationally ambiguous, "non-quantitative". If such ambiguity is not understood by the scientist, it might appear that introduction of other variables "harms" the original model. In contrast, however, modifying the original model of X by a rotation is what is desired when using the joint model: both X and Z should be fitted as well as possible. This effect does not harm the quantitative nature of the model, as long as Q value of X does not grow too much. Summary of this section: if Q computed over X elements increases significantly when modeling [X Z] instead of X, this indicates that X and Z are not compatible (when assuming this number of factors). Then analysis of [X Z] should be rejected. In all other cases, the joint model of X is equally good or better than the original model of X. If original model is rotationally ambiguous, then factors usually change: G[X Z] is different from G(X) and similarly F[Xz] is different from F(X). These new factors fit X as well as the original factors, thus they are as quantitative as the original factors. The rotation of these new factors takes into account information from matrix Z. In some cases, the new factors are rotationally unique, without any ambiguity. More often, the ambiguity of new factors is less than the original ambiguity. RESPONSE: This fullsome explanation is very valuable, and aspects of this background relevant to our paper have been added to the manuscript.

Discussion of the manuscript This manuscript suffers badly from almost complete

avoidance of equations and mathematical symbols and mathematical notation in general. Also, there are serious problems in the few equations that are present. A more compact and easier to read presentation is obtained if mathematical notation is used as the primary means of communication. It is possible that part of my criticism in this review is simply based on misunderstanding unclear and/or ambiguous verbal explanations of mathematical concepts. RESPONSE: We have addressed this by a new, more mathematical description of our methods.

The ideal of scientific work is repeatability. This ms does not provide facts that might enable repeatability, even in principle. E.g., I could not find dimensions of data matrices or obtained $Q$ values. RESPONSE: Dimensions have been added.

How were NSD data preprocessed before PMF computations? Using averages or medians? How were outliers handled? How many factors were used in each case? And so on. RESPONSE: This information has been added together with a reference to a report provided by the data provider, NPL.

The basic assumption of factor analytic modeling is that for each source, chemical profile and size distribution stay constant throughout the measurement campaign. On the other hand, it is well known that whenever nucleation happens, aerosol size distributions do vary. Also, largest particles tend to settle down more during longer transit times. In this work, constancy of size distributions was silently assumed. It might be good to discuss this fundamental question in future versions of this work. RESPONSE: Point taken. This is something that we have mentioned in previous papers with reference to the assumption that the profile of the sources does not change between emission and arrival at the receptor site.

Two-stage PMF model vs. customary PMF model In the present ms, the goal was to determine the size distributions corresponding to the previously determined aerosol composition sources. It was assumed (on what grounds?) that the rotation of the original PMF result was correct, so that the originally obtained $G$ matrix was deemed

suitable for the PMF model of NSD matrix Z. RESPONSE: We clarify the assumption that we were satisfied with the solution from the first analysis referring the reader to Beddows et al. (2015). This solution gave the best solution/rotation (agreed with all the authors during the work) to describe the urban atmosphere measured at the NK site. There are details in Beddows et al. (2015) justifying this.

In other words, it was desired that X "drive" the modeling of [X Z]. Essentially, this method corresponded to Case B, discussed above. Apparently, the authors were unaware of the one-stage method suggested for Case B. RESPONSE: We were aware of this method but avoided it in view of having a united G factor which would not be possible with a joint matrix [X Z].

In hindsight, the best approach might have been to follow both Case B and Case C, especially if there was no positive information confirming that the original PMF model of X was rotationally unique and correct. An enhanced version of the ms should briefly discuss the one-stage possibilities of doing this work according to Case B and/or Case C. RESPONSE: We have added two sections which describe Case B and case C.

The one-stage method, with suitably weighted X and Z, would be easier to explain and much easier to understand. However, it is not reasonable to expect that the work be redone using the one-stage approach. I understand step 2 so that the computed G factors from step 2 were forced to be practically identical to G factors from step 1. Is this right? RESPONSE: Yes, this is correct.

If this is right, then step 2 appears to be equivalent to non-negative weighted regression (non-negative weighted linear least squares fit) of matrix Z by columns of matrix G. This should be mentioned. RESPONSE: We have now mentioned this.

There are easy-to-use computer programs for computing such LS fits. Although PMF may also be used for this fit, using simpler tools would make the process more transparent, so avoiding unnecessary complications. Equations for defining the hidden factor should be given. The verbal definition is hard to understand and I did not manage

to understand it. RESPONSE: We hope that the clarification of the description and mathematics will mean that this is now expressed clearly.

The hidden factor, aka Nucleation factor It is a good idea to assume that due to its higher time resolution, the NSD matrix Z may contain factors that are not visible in matrix X of chemical profiles. Unfortunately, the method for defining the hidden factor(s) in Z is questionable. First of all, why did you assume that there is only one hidden factor? RESPONSE: The hidden factor was revealed as the intercept in the regression of the NSD values against the G1...G6 timeseries, after which, we then looked for it in the PMF analysis. Furthermore, from the results of Beddows et al. (2015) (for which optimum solutions were derived without factor splitting), we did not anticipate another factor to be present above 7 factors (see the Venn diagram shown in Figure 1) and this constrained our search to 7 factors, i.e. we had accounted for all of the factors in Beddows et al. (2015) and saw no need to go higher.

It seems that in stage 2, 6 factors were used. This is not defined (why not) but this is how I understand the ms. Why did you not use in 2nd stage PMF a 7th (and maybe an 8th) factor that may only fit the NSD part of the data matrix? This simple arrangement would determine hidden factor(s) avoiding the bias that non-negativity constraints may introduce in your method (see below). This alternative must be mentioned in a future version of the paper. RESPONSE: We have clarified this. We initially used 6 and then 7 to find the nucleation factor and only went to 7 factors because of the response given to the previous point, i.e. we only looked for those factors in Beddows et al. (2015).

The second Equation (3) is incorrectly formulated. Symbol j is used as a summation index on the right side. Then it cannot appear on the left side. There is a symbol "x". It is not defined, what does it mean? The text says: "The Cran R package NonLinear Minimization (nlm) (R Core Team, 2018) was used to minimise equation 3." You must not say "minimize equation". You must specify the expression that is minimized, and also specify the free variable(s) that are varied in order to minimize. I cannot understand the expression to minimize nor the free variables. For this reason, I cannot comment more

on determining the hidden factor. Maybe it is properly determined, maybe not. This part of the work is certainly not reproducible by others. RESPONSE: This equation has been correctly formulated and numbered.

Bias: It seems that the second Equation (3) is not applied to all data because of non-negativity constraints (however, there seems to be an error in the constraints, it is impossible to guess what was really intended). When some data are excluded, this creates a bias. It is impossible to know from the outside if this bias was negligible or if it distorted the results. The bias question must be documented. RESPONSE: No data has been excluded. All data was fitted with non-negative constraints.

Miscellaneous Lines 415-417 in Conclusion: "This generates confidence that the NSD and PM10 factors ascribed to one source are in fact attributable to that same source." This is a very important statement, good! There are two equations numbered (3). This caused a LOT of trouble when trying to understand the discussion of the "hidden profile" a.k.a. "nucleation profile". The first Equation 3 does not appear correctly on my computer. Possibly, it uses a symbol font that is not present on my computer so that one symbol is not visible. There is also another problem in this equation: symbol "a" is used as summation index, and symbol "a" appears also on left side. A summation index cannot be present on left side. Please check your equations before submitting new versions of the ms. Make sure that the .pdf file contains all non-standard fonts that are used e.g. in equations. RESPONSE: Changes made to clarify this matter.

The presentation should be helpful for the reader. The symbols used in text and in equations should be defined. Example: in first Eq. (3), there is symbol j. What does it mean? Is it the index of size bin? Why not help the reader and say so? In second Eq. (3), there is again a symbol j. What is it now? Please update the ms so that symbols are used in a systematic way, in order to help the reader. The following method is recommended in order to avoid confusion with symbols: RESPONSE: Changes made to provide clarity.

For your own use, create a table where each symbol, however trivial, is entered. When needing more symbols, check first with the table if the symbol is already reserved for another use. When you are ready, include short definitions from the table into the ms, either in a table of notation or to the location of first use of each symbol. Use customary matrix element notation whenever possible. In this way, you could avoid using scalar "a" first as an index and then vector a_j as a vector of unknowns. RESPONSE: This system has been adopted in the interests of clarity.

Description of the linear regression model (section 2.4) is strange. I have never seen that the coefficients are called "gradients". Also, correlations should not be mentioned when discussing linear least squares. It would be best to simply show the equation. I recommend that explanation of regression be omitted, except that the equation, using matrix element notation, should be shown. RESPONSE: Changes made as recommended.

Figure 6 is unclear. What is illustrated by the bivariate plots? Figure caption only tells that they are bivariate plots, plotted using the Openair program. Instead of naming the plotting program, it would be more important to define what is plotted vs. what, and what are the dimensions in individual diagrams. After working with the ms for a long time, I tend to guess that the "bivariate plots" might represent NSD concentrations in polar plots of wind direction and wind speed. Why did you not say this? Saving one sentence from the ms may cost hours for your new readers. RESPONSE: A description of the Bivariate plots is given as an added section and the details requested added in the figure legend.